# Interfacing Foundation Models' Embeddings

Xueyan Zou[*,§,♠], Linjie Li[*,♯], Jianfeng Wang[♯], Jianwei Yang[♯], Mingyu Ding[‡], Junyi Wei[§]

Zhengyuan Yang[♯], Feng Li[†], Hao Zhang[†], Shilong Liu[&], Arul Aravinthan[§], Yong Jae Lee[§,¶], Lijuan Wang[♯,¶]

[§] **UW-Madison**  [♯] **Microsoft**  [‡] **UC Berkeley**  [†] **HKUST**  [&] **Tsinghua University**

[¶] Equal Advisory Contribution  ♠ Main Technical Contribution  [*]Equal Contribution

https://github.com/UX-Decoder/FIND, https://github.com/UX-Decoder/vlcore

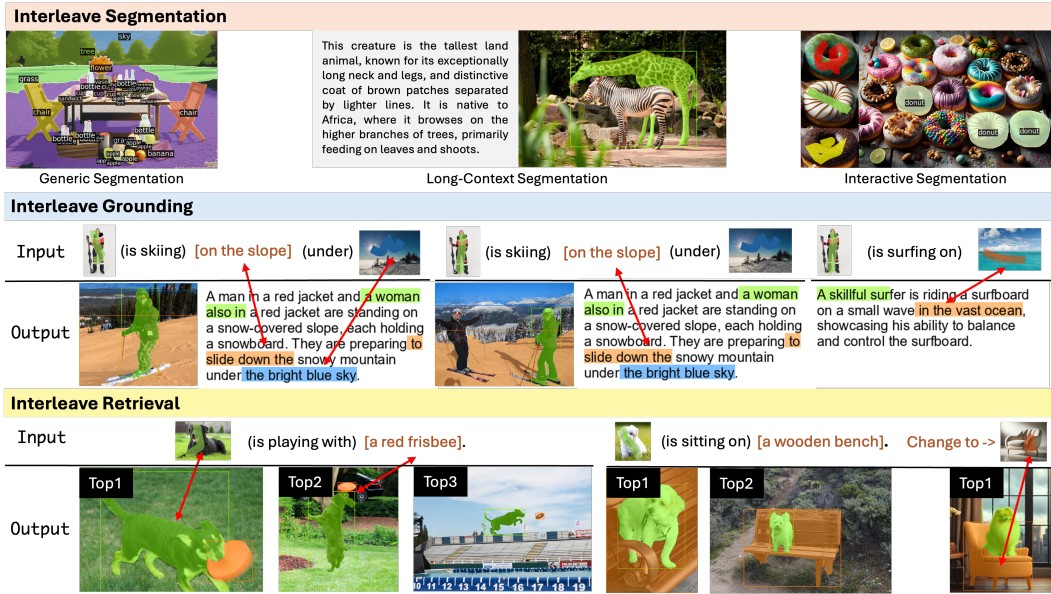

Figure 1: The proposed *FIND* interface is generalizable to tasks that span granularity (pixel to image) and modality (vision to language). The retrieval space for this figure is the COCO validation set.

## Abstract

Foundation models possess strong capabilities in reasoning and memorizing across modalities. To further unleash the power of foundation models, we present *FIND*, a generalized interface for aligning foundation models' embeddings with unified image and dataset-level understanding spanning modality and granularity. As shown in Fig. 1, a lightweight transformer interface without tuning any foundation model weights is enough for segmentation, grounding, and retrieval in an interleaved manner. The proposed interface has the following favorable attributes: (1) Generalizable. It applies to various tasks spanning retrieval, segmentation, *etc.*, under the same architecture and weights. (2) Interleavable. With the benefit of multi-task multi-modal training, the proposed interface creates an interleaved shared embedding space. (3) Extendable. The proposed interface is adaptive to new tasks, and new models. In light of the interleaved embedding space, we introduce *FIND*-Bench, which introduces new training and evaluation annotations to the COCO dataset for interleaved segmentation and retrieval. We are the first work **aligning foundations models' embeddings for interleave understanding**. Meanwhile, our approach achieves state-of-the-art performance on *FIND*-Bench and competitive performance on standard retrieval and segmentation settings.

38th Conference on Neural Information Processing Systems (NeurIPS 2024).

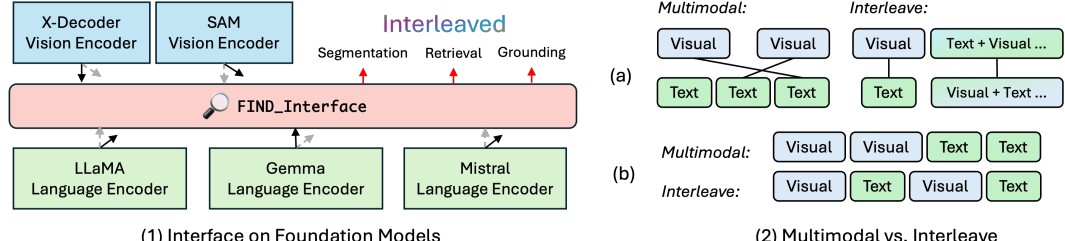

Figure 2: (1) The concept of interfacing foundation models embedding, the black arrow means active attached modules and the gray arrow means the option that it can switch to. On the right, we show the difference of Multimodal and Interleave (2.a) in the context of embeddings matching; (2.b) in the context of embeddings interaction for reasoning and generation.

# 1 Introduction

With the exhilarating progress in foundation models across the vision and language domains, such as GPT4(V) (30), DALLE-3 (31), SAM (19), and LLaMA (38), *etc.*, we have reached a stage where deep learning models achieve remarkable performances on both vision and language domains (5; 22). Specifically, models like GPT-4(V) (30) have showcased human-level perception and reasoning skills (46).

Despite their impressive capabilities in information memorization, processing, and reasoning, these models tend to be specialized for specific output types. However, their output types are limited to language for GPT, images for DALLE, masks for SAM, *etc.* In this work, we aim to leverage the privileged properties of foundation models' embeddings to expand their output space (e.g., extend to pixel-level outputs), unlocking their potential for interleaved understanding and reasoning.

To accomplish this, we introduce an INterface for Foundation models' embeDdings (FIND), which utilizes the pre-trained foundational model embeddings to jointly handle downstream tasks of varying granularities (from pixel to image) in an interleaved manner. As illustrated in Fig.2.1, the *FIND* interface processes embeddings from vision and language foundation models, and outputs segmentation, grounding, and retrieval results.

As all vision-language tasks are trained uniformly in *FIND*, an interleaved shared embedding space is created where vision and language references can be interchanged and augmented. For example, in Fig.2.2, during mapping an interleaved representation loosens the single-modality constraint on the source and target domain. And during reasoning, interleaved sequences enhance information exchange between vision and language compared to multimodal sequences.

To effectively align and evaluate the interleaved embedding space, we construct a new dataset named FIND-Bench. This dataset uses COCO images and includes new annotations for integrated grounding and segmentation. These annotations are generated by GPT-4, which, despite not processing visual input, can directly link specific image segments and annotation IDs with generated descriptions (e.g., <id>(the golden retriever) ...). This unique capability enables the creation of training and evaluation datasets for retrieval and grounding in an interleaved context.

In summary, we claim the following contributions:

- We introduce the *FIND* interface that is is generalizable, flexible, and extendable to various downstream tasks and foundation models.
- Through the effective training scheme of *FIND*, an interleaved shared embedding space is created interfacing foundation models.
- We propose a new Benchmark, *FIND*-Bench, which includes new training and evaluation ground truths for interleave segmentation and retrieval.
- Our model achieves SoTA performance on interleave retrieval and grounding and shows better or comparable performance on generic, interactive, grounded segmentation and image-text retrieval.

# 2 Related Work

**Foundation Models.** Recent years have seen a speedy evolution of foundation models in diverse areas such as computer vision (47), natural language processing (39; 10; 4; 30), and their interactions (1;

23; 44). For example, GPT-3 (4) heralds breakthroughs in natural language understanding and generation tasks, As a vision foundation model, Florence (47; 42) can be easily adapted for various computer vision tasks, such as classification, retrieval, object detection, etc.Flamingo (1) bridges powerful pre-trained vision-only and language-only models by token fusion with cross-attention. BLIP-2 (23) proposes an efficient pretraining strategy that bootstraps vision-language pre-training with a lightweight Q-Former in two stages. Different from previous multi-modal approaches, such as Flamingo (1), LLaVA (26) and Q-Former (BLIP-2) (23) that feed the vision foundation model output into a language decoder and use the LLM as an interpreter, our goal is to interface foundation model embeddings so that LLMs and vision models can be unified in the embedding space.

**Interleaved Image-Text Understanding.** Previous works have explored interleaved visual understanding in the context of visual question answering, visual dialogue, image captioning, and interleaved image retrieval (20; 13; 1). In addition, recent works (48) explore contextual detection that associates phrases with visual content in a sentence. We notice that these earlier works, though reveal interleaved capabilities for image understanding, lack an evaluation benchmark, as well as a complete training dataset. (51; 21; 2) propose a new benchmark on interleaved generation and understanding of image and document level, while there is no benchmark available for the interleaved tasks between interactive image parts and phrases. To this end, we introduce the interleaved segmentation and interleaved retrieval tasks with our carefully designed benchmark *FIND*-Bench, which we believe to be essential for the field.

**Image Understanding.** Vision Transformers (16; 37; 40; 36; 41; 12; 15; 49; 33; 34) have dominated a wide range of key image understanding tasks, such as image retrieval, detection, and segmentation. Some multimodal methods (7; 24; 50) have shown good performance for retrieval tasks. On the other hand, open-vocabulary segmentation methods have recently drawn much attention, including generic segmentation (6; 53; 11), interactive segmentation (14; 19) that separates objects by actively integrating user inputs, and grounded segmentation (53; 52) that grounds object segments from language descriptions. We notice that there is currently no available work that achieves image-level retrieval, pixel-level segmentation, and interleaved vision-language understanding in a single model. In this work, we propose *FIND* as a unified interface that can support all the above tasks, while maintaining good performance, and further enabling two new tasks of interleaved segmentation and interleaved retrieval. We unify these tasks by interfacing foundation models' embeddings.

## 3 Method

Foundation models such as CLIP (32), SAM (19), LLaMA (38), etc. can process vision or language inputs for reasoning, understanding, and generation. The embeddings generated by these models contain rich and structured information (35; 3), making them extremely well-suited for understanding tasks. Aligned with the Platonic Representation Hypothesis (17), we believe foundation models can easily communicate with each other. Therefore, we designed the FIND interface to project vision and language embeddings from foundation models into a unified space. The created space enhances both multimodal and interleaved understanding.
Since no prior benchmark exists for interleave understanding, we believe it is meaningful to formally define the interleave retrieval and segmentation problems and create a dataset for benchmarking them.

### 3.1 *FIND* Benchmark

Our new benchmark supports two tasks: interleave retrieval and interleave grounding. It evaluates both dataset-level and image-level interleave alignment, focusing on reasoning and matching capabilities. Additionally, we created training and evaluation datasets to further enhance interleave understanding.

#### 3.1.1 Task Definition

**Interleave Retrieval** [1]. An interleave entry ($E$) consists of a sequence of images (I), texts (T), and connections (C), and can be represented as $E = \langle N_1, N_2, \ldots, N_n \mid N_i \in \{I, T, C\} \rangle$, where $\langle \cdot \rangle$ is an ordered sequence. The bottom part of the Table. **??** clearly illustrates an example of an interleave entry. We denote the source domain ($\mathcal{D}_s$) of interleave retrieval as $\mathcal{D}_s = \{E_1, E_2, \ldots, E_n\}$, as shown in Fig. 3.1 (Left), and the target domain ($\mathcal{D}_t$) as $\mathcal{D}_t = \{I_1, I_2, \ldots, I_n\}$, as shown in Fig. 3.1 (Right). The task of interleave retrieval is to find the closest entry $I_* \in \mathcal{D}_t$ for each $E \in \mathcal{D}_s$, excluding itself. Formally, we define this as $\forall E \in \mathcal{D}_s, \quad I_* = \arg\max_{I \in \mathcal{D}_t, I \notin E} \mathbf{sim}(E, I)$.

---

[1] Unless we stated as interleave text retrieval, we refer to interleave visual retrieval as Fig. 3.1 shown.

Table 1: Pseudo code for Data Engine. We show the pipeline to create the *FIND*-Bench from data preparation, text prompting using GPT4, visual prompting with SEEM to integrated result.

**Interleave Grounding** [2]. An image contains a sequence of objects or segments ($O$) represented as $I = \{O_1, O_2, \ldots, O_n\}$. We provide an example of objects in the bakery image in Fig. 3.2 upper part. These objects form the target domain $\mathcal{D}_t = I = \{O_1, O_2, \ldots, O_n\}$ for interleave grounding. Unlike interleave retrieval, where interleave entries constitute the source domain, interleave grounding focuses on each component of the interleave entry, with the entities ($N$) in the interleave entry forming the source domain. Specifically, $\mathcal{D}_s = \{N_1, N_2, \ldots, N_n \mid N_i \in \{I, T\}\} \subseteq E$. We show an example of interleave entry decomposition in the lower part of Fig. 3.2. The task of interleave grounding is to find the closest entry $O_* \in \mathcal{D}_t$ for each $N \in \mathcal{D}_s$, excluding itself. Formally, we define this as $\forall N \in \mathcal{D}_s, \quad O_* = \arg\max_{O \in \mathcal{D}_t, O \notin N} \mathbf{sim}(N, O)$.

### 3.1.2 Data Engine

We reuse the images and ground truth annotations from the COCO dataset to create *FIND*-Bench. In the first part of Table. 1, we demonstrate the input data used to in-context learning for GPT-4. In addition to the COCO ground truth, we generate pseudo-image descriptions using VLM models, such as LLaVA (26), to enrich the information. In the second part of Table. 1, we present the prompt template for our data engine. This template generates the text part for the interleaved captions in part 4 of Table. 1, providing language descriptions associated with annotation IDs. The segments corresponding to these IDs are highlighted in the same color in the example image shown in Table. 1.

As stated in Sec. 3.1.1, the source and target components are exclusive. We leverage the strong visual understanding capabilities of SEEM (53) to find replacements for the visual components in the entry. The retrieved and replaced visual components are shown in part 4 of Table. 1, with the exact segment highlighted in the same color as the corresponding reference text. For example, <the playing field> is

---

[2]Unless we stated as interleave text grounding, we refer to interleave visual grounding as Fig. 3.2 shown.

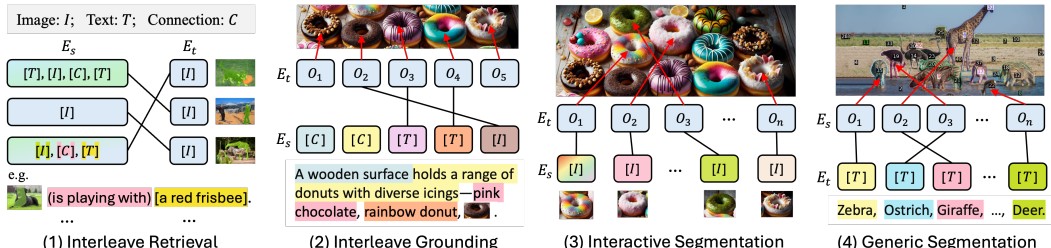

Figure 3: Task Unification for retrieval, grounding, and segmentation. The corresponding components are labeled with the same color or connected with a line or arrow.

associated with the COCO annotation ID [3171126] and a similar playing field (marked in blue) in another image. In this way, the data engine can generate comprehensive interleaved descriptions for each image in the COCO dataset. This is sufficient to build $\mathcal{D}_s$ and $\mathcal{D}_t$ for the interleave retrieval and grounding tasks introduced in Sec. 3.1.1.

## 3.2 *FIND* Approach

With benchmarks introduced in Sec. 3.1 to evaluate the model's interleaved visual understanding capability, we now present our approach for interfacing foundation models' embeddings on multimodal and interleave understanding. We begin with the preliminaries on task unification and terminology.

### 3.2.1 Preliminary

**Task Unification.** In this work, we focus on retrieval, grounding, and segmentation in both multimodal and interleaved manners. In Fig. 3, we demonstrate four example tasks: interleave retrieval, interleave grounding, interactive segmentation, and generic segmentation. From an abstract perspective, we can regard all visual understanding tasks as the problem of matching candidates from the source domain to the target domain. Formally, we define the source domain as $\mathcal{D}_s$ and the target domain as $\mathcal{D}_t$. Example elements in $\mathcal{D}_s$ or $\mathcal{D}_t$ includes interleaved entry $E$, an image $I$, an object or segment $O$, texts $T$. For each visual understanding task $\mathcal{U}(\mathcal{D}_s, \mathcal{D}_t)$, the goal is to find the closest $Y \in \mathcal{D}_t$ for each $X \in \mathcal{D}_s$. Formally we write:

$$\forall X \in \mathcal{D}_s, \quad Y^* = \arg \max_{Y \in \mathcal{D}_t} \mathbf{sim}(X, Y)$$

where $\boldsymbol{X}$, and $\boldsymbol{Y}$ are base element of $\mathcal{D}_s$, and $\mathcal{D}_t$ respectively, and $\mathbf{sim}(X, Y)$ denotes the similarity between $X$ and $Y$. For example, in generic segmentation (Fig. 3.4), $\mathcal{D}_s$ is the set of all objects (segments) in the image: $\mathcal{D}_s = \{O_1, \ldots, O_{n_s}\}$, and $\mathcal{D}_t$ is the set of category names: $\mathcal{D}_t = \{T_1, \ldots, T_n\}$. For each object $O$ in $\mathcal{D}_s$, we will find the corresponding category $T \in \mathcal{D}_t$.

**Terminology.** Here we will introduce important model terminology, including prompts ($P$) and queries ($Q$). Our model supports three kinds of inputs: vision (I), language (T), and interleaved vision-language (E). The vision and language foundation models predict the embeddings for those inputs. As shown in Fig. 4.1, by sampling the embeddings, we obtain vision prompts ($P_I$), language prompts ($P_T$), and interleave prompts ($P_E$). Additionally, trainable queries initialized with random parameters will accumulate information from the prompts. For example, in generic segmentation, object queries ($Q_O$) gather information from visual prompts. Interestingly, queries just act like "buckets" accumulating "water" (prompts) in the *FIND* interface, as shown in Fig. 4.1.

### 3.2.2 Model Pipeline

Our model is designed to interface with a pair of arbitrary vision and language foundation models. **Prompts and Queries Preparation.** Given image (I), text (T), and interleave (E) inputs, the vision encoder ($\mathbf{F}_v$) and language encoder ($\mathbf{F}_l$) will encode these inputs to sequences of embeddings $M$:

$$M_I = \mathbf{F}_v(I), \quad M_T = \mathbf{F}_l(T), \quad M_E = \{\mathbf{F}_v, \mathbf{F}_l\}(E) \qquad (1)$$

where, $M \in \mathbb{R}^{n \times d}$, and $n, d$ is the embedding number and dimension respectively. Similar to SEEM (53), we use an embedding sampler to sample customized prompts for downstream tasks. Example sampling strategies include downsampling, ROI pooling for the region, and rearrangement of embeddings for interleave prompt. The sampling procedure does not alter the embedding distribution. After sampling, we obtain $\{P_E, P_T, P_I, \ldots\} = \mathbf{Emb\_Sample}(M_I, M_T, M_E)$. Additionally, the embedding sampler is responsible for sampling queries ($\{Q_E, Q_T, Q_I, \ldots\}$) from the pool of learnable queries. We allow duplication in the sampling procedure of learnable queries. These queries

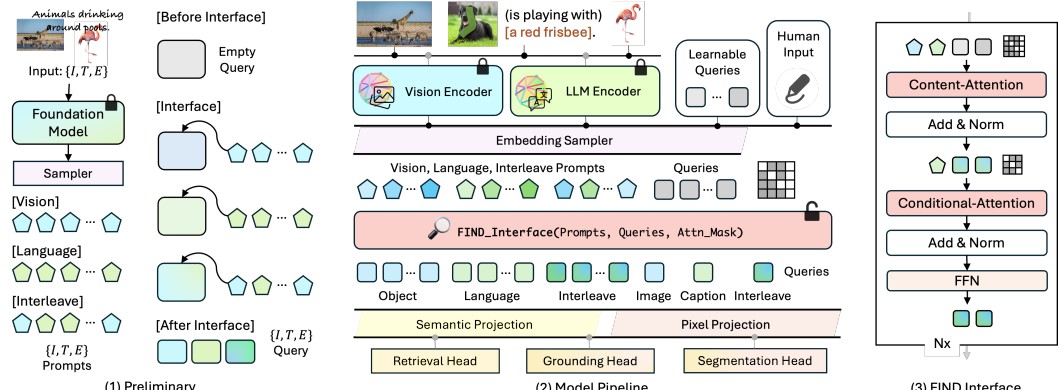

Figure 4: (a) Preliminaries on the terminology of prompts and queries. (b) *FIND* approach pipeline. The shape of different polygons represents different embedding types, and the color (vision, language) of the polygons represents input modality. (c) Detailed architecture of the *FIND* Interface.

and prompts are the inputs of *FIND* interface. Technically, the embedding sampler is usually an interpolation or grid sample layer in PyTorch.

***FIND* Interface.** The *FIND* interface primarily consists of two operations: content attention $\mathbf{A}_t$ and conditional attention $\mathbf{A}_d$, as shown in Fig. 4.3. Content attention allows queries to accumulate information from the corresponding prompts, while conditional attention enables prompts and queries to reason internally (e.g. self-attention on object queries to avoid duplication). With initial prompts $\mathbf{P}^0 = \{P_E^0, P_T^0, P_I^0, \dots\}$, and initial learnable queries $\mathbf{Q}^0 = \{Q_E^0, Q_T^0, Q_I^0, \dots\}$, content attention and conditional attention are formally defined as:

$$\mathbf{Q}^{l+1} = \mathbf{A}_t(\mathbf{P}^l, \mathbf{Q}^l; [\mathbf{P}^l \rightarrow \mathbf{Q}^l]), \quad \mathbf{Q}^{l+1}, \mathbf{P}^{l+1} = \mathbf{A}_d(\mathbf{P}^l, \mathbf{Q}^l; [\mathcal{S}^l \rightarrow \mathbf{Q}^l], [\mathbf{P}^l \rightarrow \mathbf{P}^l]) \quad (2)$$

where $\mathcal{S}^l \subseteq \{\mathbf{P}^l, \mathbf{Q}^l\}$ is a subset of queries and prompts, $\rightarrow$ represents the attention mask. For example, $[\mathbf{P} \rightarrow \mathbf{Q}]$ means that $\mathbf{Q}$ is able to attend $\mathbf{P}$ during the attention. In this way, prompts act as the information source, and queries act as the bucket. In Fig. 4.2, we unfold the prompts and queries for some tasks supported by *FIND* interface.

**Projection** The outputs of the *FIND* interface are a sequence of queries: $\mathbf{Q}^L = \{Q_O^L, Q_T^L, Q_I^L, Q_E^L, \dots\}$. We then project the queries using linear layers, $\mathbf{MLP}_s$ and $\mathbf{MLP}_p$, for semantic and pixel projection, respectively. The semantic and pixel queries are computed as $Q^s = \mathbf{MLP}_s(\mathbf{Q}^L) \in \mathbb{R}^{n_t \times d}$ and $Q^p = \mathbf{MLP}_p(\mathbf{Q}^L) \in \mathbb{R}^{n_t \times d}$, where $n_t$ is the total instance number, and $d$ is the embedding dimension. The semantic outputs are used for retrieval, category mapping, etc., while the pixel outputs are used for mask prediction.

**Task Head** With the projected queries, as illustrated Sec. 3.2.1 each understanding task can be represented as a similarity mapping procedure. Formally, segmentation result (**Mask**) can be computed given initial image embedding $M_I \in \mathbb{R}^{n_p \times d}$, where $n_p$ is the pixel number. The similarity scores (**Score**) can be computed directly from $Q^s$. The outputs for each task is a subset of {Mask, Score}.

$$\text{Mask} = Q^p \times M_I^\top \in \mathbb{R}^{n_t \times n_p}, \quad \text{Score} = Q^s \times Q^{s\top} \in \mathbb{R}^{n_t \times n_t} \quad (3)$$

**Loss** *FIND* is trained with a linear combination of losses for panoptic segmentation, grounded segmentation, interactive segmentation, image-text retrieval, interleave retrieval with visual entities from the same image, and interleave grounding. We demonstrate the loss details in the Appendix.

## 4 Experiments

**Datasets.** We use COCO (25) as our main training and evaluation dataset, which spans diverse annotation types. We make use of the annotations from COCO-panoptic, Ref-COCO (45; 28; 29), COCO-Karpathy (18), and the new datasets generated with the data engine in *FIND*-Bench. We generate two sets of new annotations, including COCO-Entity and COCO-Paragraph, the detailed statistics are shown in the table below:

| | Training | | | Evaluation | | | Entity Association | | | Average |
|---|---|---|---|---|---|---|---|---|---|---|
| | Images | Captions | Entities | Images | Captions | Entities | Mask | Phrase | Visual | Entity/Image |
| COCO-Entity | 118189 | 353219 | 1104907 | 4990 | 4990 | 15305 | ✓ | ✓ | ✓ | 4 |
| COCO-Paragraph | - | - | - | 4981 | 4981 | 22569 | ✓ | ✓ | ✓ | 7 |

**Settings.** We benchmark our method on three different model sizes: Tiny (FocalNet), Base (Davit-d3), and Large (Davit-d3). The vision backbone is fixed and reuses the X-Decoder pre-trained weights

| | Data | Joint | Generic Segmentation COCO | | | Grounded Segmentation RefCOCO-g | | COCO-Entity | | COCO-Paragraph | | Interactive Segmentation Pascal VOC | | | Image-Text Retrieval COCO-Karpathy | | COCO-Entity | | COCO-Paragraph | |
|---|---|---|---|---|---|---|---|---|---|---|---|---|---|---|---|---|---|---|---|---|
| | | | PQ | mAP | mIoU | cIoU | mIoU | cIoU | mIoU | cIoU | mIoU | Point | Circle | Box | IR@1 | TR@1 | IR@1 | TR@1 | IR@1 | TR@1 |
| *Mask2Former (T) (8) | COCO (0.12M) | - | 53.2 | 43.3 | 63.2 | - | - | - | - | - | - | - | - | - | - | - | - | - | - | - |
| *Mask2Former (B) (8) | COCO (0.12M) | - | 56.4 | 46.3 | 67.1 | - | - | - | - | - | - | - | - | - | - | - | - | - | - | - |
| *Mask2Former (L) (8) | COCO (0.12M) | - | 57.8 | 48.6 | 67.4 | - | - | - | - | - | - | - | - | - | - | - | - | - | - | - |
| Grounding-SAM (H) (27) | Grounding (5M) | ✓ | - | - | - | - | - | 58.9 | 57.7 | 56.1 | 56.6 | - | - | - | - | - | - | - | - | - |
| SAM (B) (19) | SAM (11M) | - | - | - | - | - | - | - | - | - | - | 58.2 | - | 61.8 | - | - | - | - | - | - |
| SAM (L) (19) | SAM (11M) | - | - | - | - | - | - | - | - | - | - | 68.1 | - | 63.5 | - | - | - | - | - | - |
| *SEEM (T) (53) | COCO+LVIS (0.12M) | ✗ | 50.8 | 39.7 | 62.2 | 60.9 | 65.7 | 54.3 | 56.1 | 52.6 | 54.6 | 83.5 | 86.0 | 71.8 | - | - | - | - | - | - |
| *SEEM (B) (53) | COCO+LVIS (0.12M) | ✗ | 56.1 | 46.4 | 66.3 | 65.0 | 69.6 | 57.2 | 58.7 | 56.1 | 57.4 | 87.3 | 88.8 | 75.5 | - | - | - | - | - | - |
| *SEEM (L) (53) | COCO+LVIS (0.12M) | ✗ | 57.5 | 47.7 | 67.6 | 65.6 | 70.3 | 54.8 | 57.8 | 53.8 | 56.7 | 88.5 | 89.6 | 76.5 | - | - | - | - | - | - |
| X-Decoder (T) (52) | COCO+ITP (4.12M) | ✗ | 52.6 | 41.3 | 62.4 | 59.8 | * | - | - | - | - | - | - | - | 40.7 / 49.3 | 55.0 / 66.7 | 46.5 / 52.6 | 48.0 / 55.6 | 54.8 / 62.3 | 58.5 / 66.1 |
| X-Decoder (B) (52) | COCO+ITP (4.12M) | ✗ | 56.2 | 45.8 | 66.0 | 64.5 | * | - | - | - | - | - | - | - | 50.2 / 54.5 | 66.8 / 71.2 | 49.2 / 56.9 | 51.3 / 58.1 | 58.1 / 67.5 | 62.5 / 70.1 |
| X-Decoder (L) (52) | COCO+ITP (4.12M) | ✗ | 56.9 | 46.7 | 67.5 | 64.6 | * | - | - | - | - | - | - | - | 56.4 / 58.6 | 73.1 / 76.1 | 58.1 / 60.0 | 59.9 / 62.7 | 58.7 / 71.6 | 72.0 / 74.1 |
| CLIP/ImageBind (H) (13; 9) | ITP (400M) | ✓ | - | - | - | - | - | - | - | - | - | - | - | - | 49.4 | 65.9 | 53.4 | 57.6 | 59.6 | 64.8 |
| FROMAGe (L) (20) | CC (12M) | ✗ | - | - | - | - | - | - | - | - | - | - | - | - | 27.5 | 37.8 | 27.4 | 33.1 | 32.8 | 41.3 |
| BLIP-2 (L) (23) | COCO+IPT (130.1M) | ✗ | - | - | - | - | - | - | - | - | - | - | - | - | 63.4 / 59.1 | 74.4 / 65.2 | 59.1 / 58.8 | 59.8 / 56.4 | 66.3 / 64.6 | 65.8 / 60.1 |
| FIND (T) | COCO (0.12M) | ✓ | 51.0 | 42.3 | 62.0 | 61.1 | 65.3 | 68.5 | 62.5 | 65.0 | 59.4 | 84.3 | 85.8 | 74.5 | 40.4 | 53.0 | 51.0 | 51.5 | 61.2 | 62.9 |
| FIND (B) | COCO (0.12M) | ✓ | 55.5 | 49.0 | 65.7 | 65.3 | 69.3 | 69.5 | 63.0 | 67.2 | 60.1 | 86.3 | 88.0 | 75.0 | 45.8 | 60.6 | 56.3 | 56.7 | 65.5 | 69.1 |
| FIND (L) | COCO (0.12M) | ✓ | 56.7 | 50.8 | 67.4 | 65.9 | 70.5 | 69.7 | 64.2 | 66.6 | 61.2 | 88.5 | 89.5 | 77.4 | 46.3 | 61.9 | 57.2 | 58.2 | 67.2 | 68.6 |

Table 2: Benchmark on Generalizable multi-modal understanding tasks with one model architecture joint training for all. *Unlike Mask2Former and SEEM, FIND is not trained with a deformable vision encoder. We report un-ensemble/ensemble results for X-Decoder, and the finetuned/pre-trained results for blip2. Note that we compute the ITC score for blip2 instead of ITM.

unless specified as SAM. The language backbone is a fixed LLaMA-7B, unless specified as UniCL. During training, we train the FIND-Interface jointly on all the tasks unless specified.

**Metrics.** We evaluate all the tasks with their standard evaluation metrics. For the newly proposed interleave retrieval, we use IR@5 and IR@10 (Interleave-to-image Retrieval accuracy at rank 5/10). For interleave grounding, we evaluate based on cIoU (pixel-wise IoU), and mIoU (image-wise IoU) between the predicted interleave masks and the ground truth masks.

**Baselines.** We use ImageBind (13), FROMAGe (20), BLIP2 (23) as baselines for the interleave retrieval task; Grounding-SAM (27), SEEM (53) for interleave grounding. We claim to make every effort to design the baseline evaluation protocol to achieve the best possible performance.

## 4.1 Main Results

In the main experiments, we focus on evaluating *FIND* on Generalizable, Interleavable, and Extendable capabilities as claimed in the abstract.

**(1) Generalizable to Segmentation, Grounding, and Retrieval.** Table 2 compares *FIND* with strong baselines on generic segmentation tasks including panoptic segmentation, instance segmentation, and semantic segmentation. In addition, we demonstrate the segmentation capability in both referring segmentation (RefCOCO-g: one sentence is associated with one instance) and grounded segmentation (COCO-Entity and COCO-Paragraph: one sentence is associated with multiple instances) settings. Moreover, we also benchmark *FIND*'s performance in image-text retrieval on three different ground truth types on COCO, where the average sentence length for the splits (Karpathy, Entity, and Paragraph) gradually increases. Below are the takeaways:

*The instance segmentation result stands out:* Our approach with a large vision encoder outperforms similar models like Mask2Former, X-Decoder, and SEEM, achieving a performance 2.2 points higher than Mask2Former (L), which additionally uses deformable convolution. Notably, the segmentation training data is identical for both Mask2Former and *FIND*. The performance gain likely results from our unified segmentation and grounding pipeline, which mutually benefits from the semantic ground truth of each domain.

*Mutual benefits of grounded and referring segmentation:* In *FIND*, we unify grounded and referring segmentation using queries and prompts. As shown in Table 2, our model achieves state-of-the-art performance on COCO-Entity and COCO-Paragraph and outperforms strong baselines on the Ref-COCOg dataset.

*Interactive segmentation performance is preserved in the unified settings.* Unlike SEEM which is only trained on image-only tasks, *FIND* is trained also on image-text tasks, such as image-text retrieval. With the smart design of queries, prompts, and attention mechanisms, training interactive segmentation and image-text retrieval does not interfere. Thus, it enables our approach to achieve competitive performances (i.e. *FIND* 88.5/89.5/77.4 vs. SEEM 88.5/89.6/76.5).

*Less optimal image-text retrieval results:* The sub-optimal performance in image-text retrieval is due to batch size during fine-tuning. Pilot experiments with X-Decoder showed that different resolutions (e.g., 1024 for images and 224 for language) do not generalize well across tasks. Thus, *FIND* is trained with the same resolution for all tasks. In Table 2, models are either 384x384 with batch size 384 or 1024x1024 with batch size 192 for all tasks. Other tables show results with a 640x640 training resolution and a 192 batch size.

| | | Interleave Grounding | | | | | | Interleave Retrieval | | | | Generic Segmentation | | | | | | | | |
| | | COCO-Entity | | | COCO-Paragraph | | | COCO-Entity | | COCO-Paragraph | | | Class | | Visual Context | | | Description | | |
| | cIoU | mIoU | AP50 | cIoU | mIoU | AP50 | IR@5 | IR@10 | IR@5 | TR@5 | PQ | mAP | mIoU | PQ | mAP | mIoU | PQ | mAP | mIoU |
|---|---|---|---|---|---|---|---|---|---|---|---|---|---|---|---|---|---|---|---|
| Mask2Former (L) (8) | - | - | - | - | - | - | - | - | - | - | 57.8 | 48.6 | 67.4 | - | - | - | - | - | - |
| Grounding-SAM (H) (27) | 58.9 | 57.7 | 63.2 | 56.1 | 56.6 | 62.5 | - | - | - | - | - | - | - | - | - | - | - | - | - |
| CLIP/ImageBind (H) (13; 9) | - | - | - | - | - | - | 51.4 | 61.3 | 58.7 | 68.9 | - | - | - | - | - | - | - | - | - |
| FROMAGe (L) (20) | - | - | - | - | - | - | 24.1 | 34.2 | 26.0 | 36.6 | - | - | - | - | - | - | - | - | - |
| BLIP-2 (L) (23) | - | - | - | - | - | - | 20.8 / 34.3 | 25.8 / 47.7 | 22.1 / 39.3 | 27.1 / 54.7 | - | - | - | - | - | - | - | - | - |
| X-Decoder (T) (52) | - | - | - | - | - | - | 23.6 | 32.2 | 25.6 | 35.5 | 52.6 | 41.3 | 62.4 | - | - | - | 18.5 | 15.9 | 22.5 |
| X-Decoder (B) (52) | - | - | - | - | - | - | 26.7 | 35.8 | 32.1 | 42.0 | 56.2 | 46.3 | 67.1 | - | - | - | 20.8 | 15.0 | 24.7 |
| X-Decoder (L) (52) | - | - | - | - | - | - | 26.8 | 36.2 | 32.2 | 43.4 | 57.8 | 48.6 | 67.4 | - | - | - | 23.5 | 21.1 | 21.7 |
| SEEM (T) (53) | 67.6 | 67.2 | 75.8 | 65.9 | 65.7 | 74.4 | - | - | - | - | 50.8 | 39.7 | 62.2 | - | - | - | 18.6 | 15.7 | 16.0 |
| SEEM (B) (53) | 69.4 | 69.2 | 77.8 | 69.2 | 68.6 | 77.3 | - | - | - | - | 56.1 | 46.4 | 66.3 | - | - | - | 22.9 | 21.6 | 20.0 |
| SEEM (L) (53) | 68.3 | 69.0 | 77.5 | 67.7 | 68.4 | 77.0 | - | - | - | - | 56.9 | 46.7 | 67.5 | - | - | - | 24.0 | 26.4 | 18.7 |
| FIND (T) | 74.9 | 68.1 | 79.5 | 73.2 | 66.4 | 77.7 | 43.5 | 57.1 | 49.4 | 63.9 | 51.0 | 42.3 | 62.0 | 41.8 | 32.3 | 51.6 | 19.5 | 30.2 | **35.5** |
| FIND (B) | 76.3 | 69.4 | **81.8** | **75.1** | 68.0 | 79.7 | 51.4 | 64.6 | 60.5 | 73.4 | 55.5 | 49.0 | 65.7 | 47.1 | 36.7 | 53.6 | 16.5 | 26.7 | 26.7 |
| FIND (L) | **76.3** | **69.7** | 81.7 | 74.7 | **68.6** | **79.7** | **53.4** | **66.7** | **62.7** | **75.0** | 56.7 | **50.8** | 67.4 | **49.5** | **38.9** | **57.1** | **27.0** | **31.2** | 26.8 |

Table 3: Benchmark on interleaved understanding with the jointly trained model on all tasks with one set of weights. We evaluate interleave grounding, retrieval, and generic segmentation.

| | | Generic Segmentation | | | | | | Grounding | Interactive | Retrieval | |
| | | Class | | | Description | | | g-Ref | VOC | COCO-Karpathy | |
| Vision | Language | PQ | mAP | mIoU | PQ | mAP | mIoU | cIoU | 1-IoU | IR@1 | TR@1 |
|---|---|---|---|---|---|---|---|---|---|---|---|
| X-Decoder (T) (52) | UniCL (43) | 48.5 | 39.0 | 61.4 | 12.4 | 20.7 | 18.9 | 61.3 | 82.6 | 40.4 | 54.0 |
| X-Decoder (T) (52) | LLaMa (38) | 48.5 | 38.9 | 61.2 | 19.5 | 30.2 | 35.5 | 61.6 | 82.5 | 40.2 | 52.2 |
| SAM (B) (19) | UniCL (43) | 42.5 | 37.6 | 53.6 | 4.5 | 17.7 | 17.9 | 64.9 | 81.6 | 29.1 | 39.5 |
| SAM (B) (19) | LLaMa (38) | 42.5 | 36.9 | 53.0 | 6.1 | 15.6 | 16.6 | 58.9 | 81.5 | 27.0 | 35.5 |

Table 4: Ablation study on different foundation model architectures.

**(2) Interleavable on vision and language modalities.** In Table. 3, we evaluate *FIND* on the interleaved dataset- and image-level understanding tasks in *FIND*-Bench. In the columns of COCO-Entity and COCO-Paragraph, we replace the text entity with visual reference on 0.5 probability, unlike Table. 2 the columns are purely evaluated on language-based data.

*Interleaved Segmentation:* We build an interleaved segmentation baseline using the SEEM model. Instead of formulating the grounding task in an interleaved format that SEEM doesn't support, we simply separately infer visual, and text entities using the interactive or grounding function of SEEM. As shown in Table 3, *FIND* outperforms SEEM on interleave segmentation with around +8 points on both COCO-Entity and COCO-Paragraph under cIoU metrics.

*Interleaved Retrieval:* We also explore cross-image interleave retrieval on *FIND*. Since the interleaved reference objects are from the same validation set, IR@1 is not meaningful, so we report IR@5 and IR@10 in this setting. For ImageBind and BLIP-2, we use ensemble scores of texts, sentences, and images. Following FROMAGe's settings for interleaved image-text retrieval, our performance is significantly higher than the baselines, demonstrating the effectiveness of our interleaved shared embedding space.

*Generic Segmentation:* Beyond classic evaluations using class names or fixed indices, we replace categories with class descriptions (long descriptions) or visual prompts (average features for object queries for each class). Leveraging LLMs, *FIND* excels in description-based segmentation, benefiting from smoother representations and better handling of long contexts. We also demonstrate *FIND*'s effectiveness in the visual context setting.

**(3) Extendable to arbitrary foundation models and tasks.** In the main experiments, we use X-Decoder as the vision encoder, and LLaMA as the language encoder, which shows convincing performance on all the tasks. X-Decoder has been trained to pair up vision and language embeddings, however, SAM is only trained on segmentation data without any semantic meaning. Thus, we use SAM as an ablation vision foundation model, to study how important is vision encoder trained with semantic data. For the language encoder, we adopt UniCL which has the same size as Bert to study the difference between a standard language encoder, and an LLM encoder. As shown in Table 4, UniCL and LLaMA usually have very similar performance with X-Decoder as vision encoder, except that LLaMA is extremely effective on long description reasoning. Although the performance of SAM is much worse than its counterpart X-Decoder on semantic understanding after training the interface, our approach also shows that without any modification to SAM, it applies to semantic understanding tasks on generic, grounded segmentation, and image-text retrieval.

## 4.2 Ablation Study

We ablate our approach from two perspectives: (1) What is the effectiveness of each task in the unified pipeline? (2) The effectiveness of using intermediate layers of the LLM representation.

*Independent task effectiveness:* We assess task effectiveness by gradually removing tasks in Table 5. Removing image-text retrieval significantly reduces interleave retrieval performance. Further remov-

| | | COCO | | | g-Ref | Entity | VOC | Karpathy | | Entity | |
|---|---|---|---|---|---|---|---|---|---|---|---|
| | | PQ | mAP | mIoU | cIoU | cIoU | Point | IR@1 | TR@1 | IR@1 | TR@1 |
| Task | All | 48.5 | 39.0 | **61.4** | **61.3** | 73.0 | 82.6 | **40.4** | **54.0** | **50.8** | **51.9** |
| | - Retrieval | 48.5 | 39.0 | 61.1 | 60.6 | **73.2** | **82.8** | - | - | 44.3 | 44.8 |
| | - Grounding | 48.6 | 39.1 | 61.3 | - | 40.9 | 82.8 | - | - | 45.3 | 46.2 |
| | - Interactive | 48.6 | 38.8 | 61.0 | - | 36.5 | - | - | - | 31.4 | 33.4 |
| | - Interleave | **48.9** | **39.3** | 61.0 | - | - | - | - | - | - | - |
| Language Level | [-1] | 48.3 | 39.1 | 61.2 | 61.3 | 73.0 | 82.6 | 38.9 | 52.2 | 50.3 | 50.8 |
| | [-6] | 47.8 | 38.8 | 60.4 | 60.3 | 72.9 | 81.3 | 38.1 | 49.9 | 48.1 | 47.5 |
| | [-12] | **48.5** | **39.0** | **61.4** | 61.3 | **73.0** | **82.6** | **40.4** | **54.0** | **50.8** | **51.9** |
| | [-18] | 48.2 | 39.0 | 61.1 | 62.2 | 72.6 | 82.2 | 40.1 | 52.7 | 50.6 | 50.5 |
| | [-24] | 48.5 | 38.8 | 61.5 | **61.6** | 72.9 | 82.6 | 40.2 | 52.2 | 50.5 | 51.3 |
| | [-30] | 48.1 | 39.2 | 61.1 | 60.1 | 73.3 | 82.4 | 37.9 | 49.3 | 49.4 | 50.0 |

Table 5: Ablate on each training task and language encoder feature level.

ing the grounding task decreases entity-based grounding performance. Since interleave grounding is related to interactive segmentation, removing it also reduces interleave segmentation performance. Finally, training only panoptic segmentation yields similar performance to other settings, indicating the unified interface's consistency with basic task training.

*Varying the feature embeddings layer for LLM:* LLMs process language tokens, with embeddings near input and output layers being less semantic. We hypothesize that intermediate layers align better with vision embeddings. Table 5 shows performance across tasks using emebddings from layers -1 (output) to -30 (input). Layer -12 emebddings perform best, while top and bottom layers perform worse for image-text retrieval on COCO-Karparthy splits. Thus, we use layer -12 emebddings for LLaMA throughout the paper.

### 4.3 Demonstration Results

**Interleave Album Search.** The queries in our *FIND* approach support linear complexity interleave album search. Given an image, interleave, or text input, our model can retrieve and segment all the photos in the album. Below, we show an example using the COCO validation set as the search space.

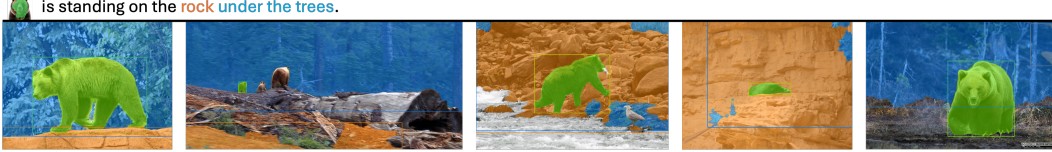

**Interleave Video Localization.** We can formulate the video frame localization problem as an image-text retrieval task. This allows us to reason about and identify corresponding objects based on given instructions, as illustrated below. We believe *FIND* is useful for robot navigation.

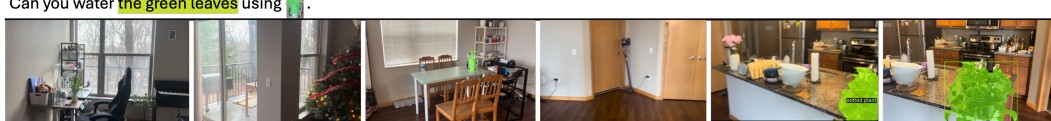

**3D Feature Field.** Foundation model embeddings are utilized to create a 3D feature field for robot manipulation, localization, and reasoning. We believe that the interleave embedding space, with its pixel-level understanding capabilities, has significant potential in the 3D feature field. Below, we compare a scene trained with FIND embeddings versus CLIP embeddings.

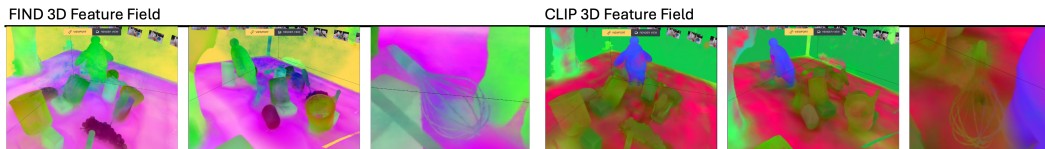

**Conclusions and Future Work.** This work introduces the *FIND* Interface, a generalized interface for aligning foundation models' embeddings, along with the *FIND* Benchmark for training and evaluation. In Sec. 4.3, we demonstrate potential applications such as interleave album search, video localization, and 3D feature fields. These examples clearly illustrate the potential of our model for personalized foundation models and robotics.

**Limitations.** Our model is only trained and evaluated on the COCO dataset. With the limitation of data quantity, we mention that the method may not be well adapted to the in-the-wild settings.

**Broader Impact.** Our proposed approach inherits ethical or social issues (e.g. bias amplification, privacy risks, energy consumption) of foundational models.

**Acknowledgement.** This work was supported in part by NSF CAREER IIS2150012, NASA 80NSSC21K0295, the Institute of Information and communications Technology Planning and Evaluation (IITP) grant funded by the Korea government (MSIT) (No. 2022-0-00871, Development of AI Autonomy and Knowledge Enhancement for AI Agent Collaboration). This research project has benefitted from the Microsoft Accelerate Foundation Models Research (AFMR) grant program.

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

# A  Method Details

## A.1  Task Specific Interface

In Section 3.2.2, we provided a comprehensive overview of the general pipeline of *FIND*. Here, we focus on the task-specific interface design choices. The pipeline comprises three main components: (1) Embeddings, which include prompts and queries as introduced in Section 3.2.2. Prompts are multimodal embeddings containing relevant information, while queries are learnable embeddings that aggregate information from the prompts. For instance, for image prompts (a.k.a visual features of an image) we denote them as `p.image`. (2) Operators, which incorporate both content and condition attention, and are responsible for information accumulation and exchange. The arrows $\leftarrow$, $\leftrightarrow$ denote the attention direction. (3) Projection, which maps the queries into semantic or pixel space. Table. 6 below shows details of all task-specific design choices for the *FIND* interface, including embeddings, operators, and projection.

| Task | Tokenization | | Reasoning | | Detokenization |
|------|-----------|---------|-------------------|---------------------|------------|
| | Prompts | Queries | Content Attention | Condition Attention | Projection |
| Generic Segmentation | image, class | object, class | q.object ← p.image
q.class ← p.class | p.* ↔ p.*, q.* ↔ q.* | Pixel, Semantic |
| Grounded Segmentation | image, image, text | grounding, text | q.grounding ← p.image
q.text ← p.text | p.* ↔ p.*, q.* ↔ q.*
q.grounding ← p.text | Pixel, Semantic |
| Image-Text Retrieval | image, caption | image, caption | q.image ← p.image
q.caption ← p.caption | p.* ↔ p.*, q.* ↔ q.* | Semantic |
| Interactive Segmentation | image, spatial | segment, spatial | q.segment ← p.image
q.spatial ← p.spatial | p.* ↔ p.*, q.* ↔ q.*
q.segment ← p.spatial | Pixel, Semantic |
| Interleave Grounding | image, interleave | entity, interleave | q.entity ← p.image
q.interleave ← p.interleave | p.* ↔ p.*, q.* ↔ q.*
q.entity ← p.interleave | Pixel, Semantic |
| Interleave Retrieval | image, interleave | image, _interleave | q.image ← p.image
q._interleave ← p.interleave | p.* ↔ p.*, q.* ↔ q.* | Semantic |

Table 6: Task specific *FIND* Interface. We define each task under the prototype of the *FIND* interface that enables a shared embedding space, and a unified and flexible architecture for future tasks. Where $p$, $q$ stands for prompts, queries, and arrows stand for attention direction. The colors red, blue, and olive are the embeddings of vision, language, and interleave modality.

## A.2  Loss Functions

The training tasks include panoptic segmentation, interactive segmentation, grounded segmentation, image-text retrieval, interleave retrieval with visual entities from the same image, and interleave grounding. Losses for each task are standardized loss functions including $\mathcal{L}_{\text{BCE}}$ for binary cross-entropy loss, $\mathcal{L}_{\text{CE}}$ for cross-entropy loss, $\mathcal{L}_{\text{DICE}}$ for dice loss, $\mathcal{L}_{\text{C}}$ for contrastive loss. Below is the loss function for *FIND*.

$$\mathcal{L} = \alpha_p \mathcal{L}_{\text{CE\_pano}} + \beta_p \mathcal{L}_{\text{BCE\_pano}} + \gamma_p \mathcal{L}_{\text{DICE\_pano}} + \alpha_g \mathcal{L}_{\text{CE\_grd}} + \beta_g \mathcal{L}_{\text{BCE\_grd}} + \gamma_g \mathcal{L}_{\text{DICE\_grd}}$$
$$+ \alpha_i \mathcal{L}_{\text{CE\_iseg}} + \beta_i \mathcal{L}_{\text{BCE\_iseg}} + \gamma_i \mathcal{L}_{\text{DICE\_iseg}} + \theta \mathcal{L}_{\text{VLC\_imgtexr}} + \phi \mathcal{L}_{\text{IC\_intr}} + \alpha_{ig} \mathcal{L}_{\text{CE\_intg}} \quad (4)$$
$$+ \beta_{ig} \mathcal{L}_{\text{DICE\_intg}} + \gamma_{ig} \mathcal{L}_{\text{ICE\_intg}}$$

where, `pano` denotes panoptic segmentation, `grd` denotes grounding, `iseg` denotes interactive segmentation, `imgtextr` denotes image-text retrieval, `intr` denotes interleave retrieval, `intg` denotes interleave grounding. For more implementation details on the loss function, please refer to the code.

## A.3  Case Study: Interleave Grounding

As shown in Table. 6, the input embeddings of interleave groundings for *FIND* interface contain prompts and queries. Image prompts are the image features with a shape of $[h \times w, 512]$, while interleaved prompts are visual-language tokens of sentences like "A baseball player in a black and white uniform crouches on ▨ near ▨ holding a ▨ taking a break." with a shape of $[l, 512]$ ($l$ is the token length). Entity queries are learnable embeddings for object proposals of the image, shaped $[100, 512]$. Interleave queries are learnable embeddings for gathering information from the interleave prompts, shaped $[n, 512]$, where n is the total number of meaningful entities. For example, the interleave sentence shown above has entity numbers of 4. Specifically the entity contains ['A

baseball player in a black and white uniform, , , ], which is the total number of $[T]$ and $[I]$ referencing to Fig. 3.

After getting a full sense of the input embeddings of interleave grounding, including `p.image`, `p.interleave`, `q.entity`, `q.interleave`. We then introduce the operation on top of those embeddings. As introduced in Sec. 3.2.2, the operations contain content attention $\mathbf{A}_t$ and conditional attention $\mathbf{A}_d$. Formally we could write the attention mechanism for the specific input embeddings of interleave grounding with the following equations:

$$\texttt{q.entity, q.interleave} = \mathbf{A}_t([\texttt{q.entity,q.interleave}]; [\texttt{p.image,p.interleave}]; \mathbf{M}_t), \tag{5}$$

$$\texttt{q.*, p.*} = \mathbf{A}_t([\texttt{q.*, p.*}]; [\texttt{q.*, p.*}]; \mathbf{M}_d) \tag{6}$$

where $\mathbf{A}(\texttt{query}; \texttt{key=value}; \mathbf{M})$ is the attention operator with query, key, value and mask. Given the order `p.image`, `p.interleave`, `q.entity`, `q.interleave`, the content and condition attention masks are written below:

$$\mathbf{M}_t = \begin{bmatrix} F & F & F & F \\ F & F & F & F \\ \mathbf{T} & F & F & F \\ F & \mathbf{T} & F & F \end{bmatrix} \mathbf{M}_d = \begin{bmatrix} F & F & F & F \\ F & \mathbf{T} & F & F \\ \mathbf{T} & F & \mathbf{T} & F \\ F & \mathbf{T} & F & \mathbf{T} \end{bmatrix} \tag{7}$$

The index of matrix coordinates follows the input order. After the input prompts and queries are fully communicated, we will compute the projected pixel and semantic embeddings for output in the following manner:

$$\texttt{q.entity}^s, \texttt{q.interleave}^s = \mathbf{MLP}_s(\texttt{q.entity, q.interleave}) \tag{8}$$
$$\texttt{q.entity}^p = \mathbf{MLP}_p(\texttt{q.entity}) \tag{9}$$

where $^s,^p$ are semantic and pixel projection respectively. This way, queries are projected into semantic and pixel space to compute the final output. The dimension of $\texttt{q.entity}^s$ and $\texttt{q.entity}^p$ are both $[100, 512]$. In addition, $\texttt{q.interleave}^s$ has dimension $[n, 512]$ where n is the entity number. With those projected queries and image features $M_I$ in the pixel projection space with shape $[h, w, 512]$. We could get the final output mask associated with each entity with the following operation:

$$\texttt{Index} = \arg \max_{\texttt{dim=0}} \mathbf{sim}(\texttt{q.entity}^s, \texttt{q.interleave}^s) \tag{10}$$
$$Q_p^* = \texttt{q.entity}^p[\texttt{Index}] \tag{11}$$
$$\texttt{Mask} = Q_p^* \times M_I \tag{12}$$
$$\tag{13}$$

In this way, we associate the grounding entity with the desired mask segment of the image, as shown in the top right figure in Table. 1.

