# OpenReview forum: "Interfacing Foundation Models' Embeddings"
_NeurIPS.cc/2024/Conference — NeurIPS 2024 poster_

### Official Review · Reviewer_AFHH · 2024-06-28

**Soundness:** 3
**Presentation:** 1
**Contribution:** 2
**Rating:** 3
**Confidence:** 5

**Summary:**

The paper proposes to construct an interface to connect the embeddings and predictions from different foundation models. With the designed interface, the overall system has a promise to interleave any modality in a flexible manner. To showcase this flexibility, this paper constructs a benchmark, FIND, by leveraging existing COCO labels and text-only GPT-4 model. As a generalist model, the paper shows that FIND generally performs better than existing generalist model and sometimes even specialized model.

**Strengths:**

- The paper idea is novel and promising if extended well.
- The paper conducts an extensive experiments on COCO and compares with valid baselines.

**Weaknesses:**

Method and experiment
- Lack of baseline of instruction-tuned vision-language models, such as LLaVA.
- Lack of out-of-domain evaluation. The advantages of foundation models is that they have a higher possibility to generalize to unseen domain due to their scale. However, it is unclear when the COCO-trained interface is able to generalize to other domains.

Writing to improve
- $\textbf{sim}$ is first introduced in L109 and is later explained in L149
- Figure 3 is not self-contained. It’s hard to follow what E, O mean. There are too many notations and too little text to understand the tasks and how the task unification works here.
- In Figure 4 (center), the caption says “the shape of different polygons represents different embedding types” I cannot find where in the text specifying different “types of embeddings”.
- Missing citations in L206,207 after each model
- Where is “interleaved segmentation” in Table 3 mentioned in L256

**Questions:**

Benchmark
- From Table 1, it is not clear to me how does the text-only data engine resolve the cases where there are multiple instances within the same classes. For example, in `Prompt for GPT4 Engine`, say the image contains ten mugs on a table. How does data engine resolve the mapping between the bounding boxes of each mug to the captions?

Method
- How does the embedding sampler work for Llama? Specially, which embeddings is used? Is it the last hidden states of the last tokens? Do the authors have any insights in extracting the embeddings from an auto-regressive model? It is non-trivial to use the embeddings in an auto-regressive mdoels, this paper [1] specifically ablate the design choice.
- Does the approach involves any similarity calculation? In section 3.2.1 task unification, the authors use a similarity measurement to connect the source and the target domains. While in the Fig. 4, I feel the FIND interface is purely relied on a transformer network followed by several projection heads. Can the authors clarify the method?


Experiments
- In Table 2
  - Why SEEM, X-decoder, BLIP-2 is marked as not “jointly trained” (the third column). Please clarify it.
  - What does * mean? It’s shown in the first column and the seventh column (mIoU for RefCOCO-g).
- In section 4.1, the authors claim that the less optimal image-text retrieval results are due to the batch size during fine-tuning. In L245-248, I am very confused with the justification. Here are my questions:
  - What resolution FIND trained with?
  - “Pilot experiments with X-decoder showed … well across tasks.” Is this described in X-decode paper or is it reproduced by the authors? Also, I checked the section 4.1 in X-decoder paper, I believe the authors want to say “1024 for segmentation and 224 for image-text data”.
  - I am not sure if the authors are claiming that the main issue is the batch size or the resolutions. If it’s the batch size, why not use gradient accumulation?


[1] Vision-Language Models Provide Promptable Representations for Reinforcement Learning

**Limitations:**

The limitations include
1. Presentation is not clear enough and has a lot to clarify
2. The paper works around foundation models but it's limited to COCO dataset.
3. The benchmark data engine requires more qualitative examples and justification of the design choices.

---

> ### Author Rebuttal · Authors · 2024-08-07
>
> **We sincerely appreciate your thorough and comprehensive reading of our paper.** We understand your frustration with the confusing implementation details despite your diligent effort. We apologize for any confusion caused. **We kindly ask that you consider the paper from a higher-level perspective**, as highlighted by the strengths noted by Reviewer-kGcR and Reviewer-iZKM. Meanwhile, we will address these constructive detailed comments in the camera-ready version if any.
>
> [Q1] Lack of Instruction baseline e.g. LLaVA.
>
> In **[Common Question 1]**, we provide a comprehensive discussion on why instruction-based methods are not comparable to our approach, considering both the output type and the specific problem we are addressing. Additionally, in **[Common Question 3]**, we thoroughly explain the importance of multimodal understanding and mapping in the steerable era.
>
> [Q2] Lack of out-of-domain evaluation.
>
> Thank you for pointing this out. We have created a new benchmark on SAMv2, using an improved protocol outlined in Table.1 for interleave grounding (we have illustrated this improved protocol in the rebuttal PDF). This benchmark is compared with SoM+GPT-4o, which serves as a high-standard evaluation protocol, the SoM mark is labeled with our best power SEEM model.
>
> → SAMv2 Benchmark statistics.
>
> | Dataset | # Images (videos) | # Instances | Avg Text Length |
> | --- | --- | --- | --- |
> | sav_val | 136 | 532 | 221 |
> | sav_test | 116 | 451 | 209 |
>
> → SAMv2 Experiment results.
>
> |  | sav_val |  |  | sav_test |  |  |  |
> | --- | --- | --- | --- | --- | --- | --- | --- |
> | Model | mIoU | cIoU | AP50 | mIoU | cIoU | AP50 | Latency |
> | FIND_Focal-T_LLaMA_640 | 58.9 | 63.7 | 65.6 | 57.8 | 59.4 | 66.5 | 0.0809 s/iter |
> | FIND_Focal-T_UniCL_640 | 58.3 | 65.7 | 65.9 | 60.1 | 60.3 | 68.7 | 0.0528 s/iter |
> | FIND_Davit-d5_LLaMA_640 | 62.5 | 69.8 | 71.6 | 60.2 | 64.6 | 71.3 | 0.1978 s/iter |
> | FIND_Davit-d5_UniCL_640 | 61.7 | 66.0 | 72.3 | 62.0 | 65.6 | 72.2 | 0.1699 s/iter |
> | SoM + GPT4o | 75.3 | 83.6 | 75.3 | 76.7 | 81.2 | 76.7 | ~10 s/iter |
>
> [Q3] Sim is first introduced in L109 and is later explained in L149.
>
> We apologize for the confusion. We will add a reference the first time we use the term “sim.” Initially, we did not include it because “sim” seemed straightforward to understand.
>
> [Q4] Hard to follow what E, O mean in Fig.3.
>
> We have clearly defined E in L103, and O L110 of the main paper.
>
> [Q5] Too many notations and too little text to understand the tasks and how the task unification works here.
>
> The mathematical symbols are carefully curated to abstract and unify the tasks, as highlighted by Reviewer-kGcR in strength 3: “The writing and visualizations are clear and well-presented.” We provide a comprehensive textual explanation of task unification in the rebuttal PDF, section xxx.
>
> [Q6] Cannot find where specifying different “types of embeddings” in Fig.4.
>
> The types of prompts and queries are clearly labeled in Fig. 4. You can find the text “Vision, Language, Interleave Prompts” directly above the pentagon (5-edge-polygon), and “Queries” above the gray square.
>
> [Q7] Missing citations in L206-207 after each model.
>
> Thanks so much for pointing this out, we will add the citation in the camera-ready version if any.
>
> [Q8] What is interleaved segmentation in Table.3.
>
> Thanks for pointing out, Interleave Segmentation actually means Interleave Grounding defined in L110. We will edit this in Camera Ready if any.
>
> [Q9] How does the text-only data engine resolve the cases where there are multiple instances within the same classes?
>
> - In Table. 1 **SI** contains object descripts generated by LLaVA, together with bounding box and annotation ID, the GPT4 data engine is able to identify the object appearance. We show an example to resolve the confusion on “ten-mug problem” in the section [Qualitative examples on FIND-COCO-Bench] of the **rebuttal PDF**.
> - In addition, for the newly created SAMv2 benchmark, we are using SoM [1] + GPT4-V to further reduce object hallucination. We show 7 random examples for FIND-SAMv2-Bench in the **rebuttal PDF**.
>
> [Q10] How does the embedding sampler work for Llama.
>
> For language embedding, we use identity sampling for all language embeddings. Please refer to [Common Question 2] for more details.
>
> [Q11] Any insights into extracting the embeddings from an auto-regressive model?
>
> Yes, this is one of our main contributions, clearly stated in L49-50 and L66. Additionally, Table 5 shows that the -12 layers feature in LLaMA aligns best with visual representation in terms of semantic meaning. We have demonstrated this insight in L290-295.
>
> [Q12] Related work “Vision-Language Models Provide Promptable Representations for Reinforcement Learning”.
>
> - The related work uses the last layer embeddings from LLMs as a knowledge prior for policy generation. They ablate the design choices of “No Prompt; No Generation; Change Aux.; Text Oracle Detector” in Table 8, which are very different in content and objective from our approach.
> - The referenced paper is a concurrent work with the NeurIPS template, updated in May 2024 on ArXiv.
>
> [Q13] Does the approach involve any similarity calculation?
>
> Task unification is achieved by treating all tasks as similarity mapping problems. This similarity mapping is applied after “the transformer network and several projection heads”; it is used to compute the loss and identify the output. We have provided a use case study in the rebuttal PDF, Sec.xxx, demonstrating how similarity mapping is applied to interleave grounding.
>
> [Q14] What is jointly trained?
>
> Joint training means whether the model is jointly trained on all the tasks with the number indicated in Table.2. We apologize for the typo that X-Decoder and SEEM should be jointly trained.
>
> [Q15] What does * mean in Table.2 for mIoU?
>
> We apologize for the confusion. * means the model can evaluate this number, but it was not reported in the original paper, so we did not include it.

---

> ### Author Response · Authors · 2024-08-07
>
> [Q16] What resolution FIND trained with?
>
> We have clearly stated in L245-248:
>
> *“In Table 2, models are either 384x384 with batch size 384 or 1024x1024 with batch size 192 for all tasks. Other tables show results with a 640x640 training resolution and a 192-batch size.”*
>
> [Q17] “Pilot experiments with X-decoder showed … well across tasks.” Is this described in X-decode paper or is it reproduced by the authors?
>
> The experiments are produced by the authors.
>
> [Q18] Presentation is not clear.
>
> We hope you have a better understanding after the rebuttal and will update the camera-ready version if needed. As Reviewer-kGcR noted, “The writing and visualizations are clear and well-presented.”
>
> [Q19] Limitation to COCO dataset.
>
> Thank you for the suggestion. We have addressed this by introducing the new FIND-SAMv2-Bench. Please refer to [Q2] for more details.
>
> [Q20] Need more qualitative examples for the data engine and justifications.
>
> We have provided additional justifications for the design in our response to [Q9], and more qualitative examples are included in the **rebuttal PDF**.

---

> ### Comment · Reviewer_AFHH · 2024-08-09
>
> Thanks for the detailed clarification.
>
> I read through the rebuttal. Overall, I start to appreciate the idea of the proposed framework *in high-level*, but I need some time to put the submission and the clarification in the rebuttal together to refine my understanding.
>
> To facilitate the discussion, I want to start by adding a follow-up question:
> 1. I appreciate the effort for the new dataset [Q2]. Can the author give some interpretations of the provided table?

---

> > ### Author Response · Authors · 2024-08-13
> >
> > We again appreciate your review **actually helped us rethink our model in a more logical way** than ever, and we hope ***the following contents could facilitate your understanding towards the final decision*** because we do believe our work is *novel and inspiring for future works.* We also aligned your question with the structure provided below.
> >
> > > **High-Level Motivation**
> > >
> > 1. We aim to align embeddings across different granularities (e.g., segments and images for visual understanding; paragraphs and phrases for language) within foundation models like X-Decoder and LLaMA.
> > 2. We seek to align embeddings between foundation models across modalities (e.g., Vision and Language), consistent with the Platonic representation hypothesis.
> > 3. We aim for the aligned embeddings to communicate seamlessly and efficiently.
> > 4. We want to identify the optimal design choices for embedding extraction from different foundation models.
> > 5. We believe all understanding tasks fall under the broader scope of information mapping. → [Q3, Q13]
> >
> > > **Interface Design aligned with Motivation**
> > >
> > 1. We design the interface to incorporate queries and prompts that unify granularity within a foundation model. Specifically, queries can attend to an arbitrary number of prompts through content attention, enabling unified granularity embeddings. → [Q6, Q10]
> > 2. Additionally, since queries can attend to prompts across different modalities through content attention, we align these modalities within the same representation space via queries. → [Q6, Q10]
> > 3. The condition attention allows queries that span granularity and modality to communicate efficiently. → [Q4]
> > 4. We utilize object queries for vision embeddings and exhaustively test which layer embeddings are optimal for language models like LLaMA. → [Q11]
> > 5. Projection and similarity mappings enable the unification of all the recognition tasks under a consistent framework. → [Q5, Q13]
> >
> > > **Experiments proves the Design choice and Motivation**
> > >
> >
> > We evaluated generic segmentation, interactive segmentation (Table 2, Main Paper), and text-paragraph grounding (Fig. 1, Second Row, Main Paper) to demonstrate unification across granularity within the foundation model, in both vision and language.
> >
> > 1. We assessed grounded segmentation and image-text retrieval (Table 2, Main Paper) to validate cross-modality alignment (vision and language). Additionally, we confirmed that this alignment is invariant across foundation models trained independently (Table 4, Main Paper).
> > 2. To evaluate the effectiveness of cross-granularity and cross-modality communication, we developed the FIND-Bench for interleave grounding and retrieval. FIND-Bench is compared with other models focused on information fusion in Table 3. → [Q2, Q9]
> > 3. The design choices for vision have been explored in previous works (e.g., Mask2Former, X-Decoder, SAM). We conducted an ablation study on language design choices in Table 5 (Main Paper, Lower Part) to determine how to extract the correct information from language models.
> > 4. In Table 5 (Upper Part), by gradually removing each task in FIND, we demonstrated the effectiveness of similarity mapping on recognition tasks.
> >
> > > **A Missing Piece for Alignment**
> > >
> >
> > We believe that embeddings from different models within the same modality can also effectively communicate. Several concurrent pilot studies, such as [1, 2, 3], are exploring this direction.
> >
> > > **Future Research**
> > >
> >
> > On the road towards the most powerful “GPT4-V”, there are three major contents for the final models:
> >
> > 1. Tokenization across modality.
> > 2. Effective communication and reasoning between modalities.
> > 3. Detokenized to human language.
> >
> > Our model focuses on the effective communication and reasoning between modalities (**Point 2**), where these components should work together seamlessly. In the multimodal section (7.2) of LLaMA 3.1 [4]. their approach is similar to ours in terms of design choices, but we extend this concept to a finer granularity (pixel-image, phrase-paragraph). **Our approach is novel in this direction, addressing an area that is essential yet underexplored**.
> >
> > For approaches focusing on **instructional tuning** (e.g., LLaVA), **they primarily work on Point 3**, detokenizing into human language for steerable interaction. ***We emphasize that our exploration is a parallel effort, addressing different aspects of the problem.*** → [Q1]
> >
> >
> >
> > [1] Cambrian-1: A Fully Open, Vision-Centric Exploration of Multimodal LLMs
> >
> > [2] MouSi: Poly-Visual-Expert Vision-Language Models
> >
> > [3] Theia: Distilling Diverse Vision Foundation Models for Robot Learning
> >
> > [4] The Llama 3 Herd of Models

---

> ### Author Response · Authors · 2024-08-09
>
> Thanks so much for your reply, **feel free to ask any further questions if needed** : )
>
> For the additional table in [Q2], we leverage the newly released SAMv2 [1] dataset that originally worked on video object/part segmentation. We provide a link [2] below for your reference on the dataset explorer. The dataset contains one validation set, and one test set, we generate new benchmark labels for those videos. SAMv2 (a.k.a SAV) dataset has the following benefits in solving your confusion:
>
> (1) Video datasets are definitely in another domain compared with the COCO dataset, e.g. they contain motion blur and the scene is usually different from COCO-style scenes. This will solve your confusion on [Q2] better.
>
> (2) Benchmarking on the video dataset will unleash more potential for our model, we have observed qualitatively that FIND can do interleave video tracking on SAMv2 dataset. If there is a future version, we will add those results (the rebuttal cannot add figures or links).
>
> -> We create the FIND-SAV-Bench using the following protocol:
>
> (1) We generate instance-level annotations using SEEM-d5 [3] and filter out the annotations with (a) Low confidence. (b) Very small region.
>
> (2) We use Set-of-Mark prompting [4] to annotate the visual images with the generated annotations. This will give better results than purely using language prompts as shown in Table.1 (Main paper).
>
> (3) We prompt GPT4-V with the following exact prompt, this is an improved version of the method shown in Table.1 in solving the lack of fine-grained annotation in SAMv2:
>
> ```
> 1. Can you describe each instance in the [image1] with detailed appearance, in the order of how confident you are on the recognition? \n
>
> 2. Can you selecte 3-5 instances in the [image1] that also likely appear in the [image2]? \n
> After these steps, generate image captions with grounded entities and attributes with following instructions: \n
> 1. [image1] is the sampled image to generated grounded caption. \n
> 2. [image2] is the reference image help to select which entities in [image1] should be included in the generated caption. \n
> 3. Numbered instances in [image1] are the proposed candidate entities. \n
> 5. The number in [image1] and [image2] are different, you should use in the number in [image1]. \n
> an example output format would be: ##output##:"[entity_id]<A woman> sitting next to [entity_id]<a handsome man>, with their hands holding together under [entity_id]<the blue sky>.", where [entity_id] and <xxx> are associated with the ground truth bounding boxes. \n
> generated caption constraints:
> 1. [entity_id] should be the same as the number in [image1], e,g: [1]. \n
> 2. Try to find the entities in [image1] that also appear in [image2]. \n
> 3. The selected entity should be the instance you are very confident on recoginition, selecte around 3-4 entities would be fine. \n
> 4. The entity description in <> should be accurate and detailed to identify the instance, e.g. <the dog with black and white dot>, <the second bottle from the front>. \n
> 5. Focus more on instance classes instead of stuff classes. \n
> Please generate the grounded caption for [image1] accordingly.\n
> ```
>
> (4) We do human pruning (approve/decline) to remove any ridiculous examples.
>
> The final examples of FIND-SAMv2-Bench are shown in the **rebuttal PDF** with the section name "Qualitative examples on FIND-SAMv2-Bench". And the statistics are shown in the [SAMv2 Benchmark statistics], with the metrics below:
>
> (1) # Images (videos): The total number of videos/images in the splits, because we only annotated the first image in a video for benchmarking as the frame looks similar inside one video. This labeling strategy will maximize evaluation capability with minimum effort.
>
> (2) # Instances: Total number of instances annotated in the split.
>
> (3) Avg Text Length: The average sentence length of the grounding annotation in character.
>
>
> [1] Ravi, Nikhila, et al. "SAM 2: Segment Anything in Images and Videos." arXiv preprint arXiv:2408.00714 (2024).
>
> [2] https://ai.meta.com/datasets/segment-anything-video/
>
> [3] Zou, Xueyan, et al. "Segment everything everywhere all at once." Advances in Neural Information Processing Systems 36 (2024).
>
> [4] Yang, Jianwei, et al. "Set-of-mark prompting unleashes extraordinary visual grounding in gpt-4v." arXiv preprint arXiv:2310.11441 (2023).

---

> ### Author Response · Authors · 2024-08-09
>
> After the FIND-SAMv2-Bench is prepared, we evaluate the interleave grounding with the FIND model. Because the video/frame number is too few in SAV val and test sets we do not propose interleave retrieval evaluation here.
>
> For interleave grounding, we use the following protocol to generate the query: For each entity we want to do the grounding, 0.5 probability is applied to determine whether we want to use visual or text reference for grounding. An example grounding example is shown below:
> ```
> [1]<A woman> sitting next to [2]<a handsome man>, with their hands holding together under [3]<the blue sky>.
> ```
> Thus, <> would either be text or visual reference of the instance. The visual reference is the masked scribble part of the original instance in the image, one better option could be the tracked instance in another frame.
>
> We evaluate the FIND-SAMv2-Bench on our Tiny and Large (d5) model with both UniCL (CLIP style language encoder) and LLaMA language encoder. As shown in the table [SAMv2 Experiment results], in addition to the effective interleave grounding in the out-of-domain dataset, we have also observed that our interface is more effective when the vision backbone is stronger (e.g. davit-d5-Florence) with evidence on the comparison results between LLaMA or UniCL language encoder.
>
> To further prove the effectiveness of our FIND approach, we compared it with the strongest multimodal baseline of GPT4-o, as it is not able to do interleave grounding from scratch, we use SoM [1] as the adapter to bridge pixel-level visual grounding in GPT4-o. It is clearly shown in the table [SAMv2 Experiment results], that FIND similar results on AP50 metrics with GPT4-o while 58 times faster than GPT4-o.
>
> The exact prompt for evaluating GPT4-o are shown below:
> ```
> You are given a marked image with masks and number in each region.
> Given the full sentence {}, and the corresponding viusal reference, you are instructed to select the region that best matches the query entities {}.
> The output format should be: ##output##: [Entity_id]: (Region number), [Entity_id]: (Region number), ...
> ```
>
> [1] Yang, Jianwei, et al. "Set-of-mark prompting unleashes extraordinary visual grounding in gpt-4v." arXiv preprint arXiv:2310.11441 (2023).

---

> ### Author Response · Authors · 2024-08-14
>
> Dear Reviewer AFHH,
>
> There are only four hours left before the discussion deadline, it would be appreciated if the reviewer could take a look at the answers and give the final decision! We again appreciate your reviewing efforts : )
>
> Best,
> Authors

---

### Official Review · Reviewer_kpHd · 2024-07-06

**Soundness:** 2
**Presentation:** 2
**Contribution:** 2
**Rating:** 5
**Confidence:** 4

**Summary:**

FIND is a generalized interface for aligning foundation models' embeddings using a lightweight transformer without tuning pretrained model weights. It supports various tasks like retrieval and segmentation, is adaptable to new tasks and models, and creates a shared embedding space through multi-task training. FIND-Bench, an extension of the COCO dataset, showcases its effectiveness, achieving state-of-the-art performance in interleave segmentation and retrieval.

**Strengths:**

1. The starting point and motivation of the paper is good.
2. The experimental results are very good.

**Weaknesses:**

1. This paper is somewhat difficult to understand. For example, what is the **embedding**? Is it the embedding generated by the tokenizer or the feature generated by the foundation model? And what does **interleaved** mean? My understanding is that different modalities are interleaved. It is not limited to image+text or text+text; it can be image+text+image+text. I am not sure if my understanding is correct. However, it took me many careful readings to figure this out. It would be best to clearly define and emphasize key terms and settings at the very beginning.
2. The method in section 3.2.2 is somewhat simplistic and lacks innovation. It doesn't seem to have much new design. It needs to be clarified why this interleaved approach is challenging (for example, if all the inputs are just treated as a sequence, will the performance be significantly worse? If so, why?).  More analysis and intuition about that will be better.

**Questions:**

Please see Weakness part.

**Limitations:**

No Limitations.

---

> ### Author Rebuttal · Authors · 2024-08-07
>
> **We greatly appreciate your comprehensive comments in the weakness section. We sincerely hope that, after reading our rebuttal, you will have a new perspective.** We believe most of the confusion stems from terminology common within the small multimodal understanding community. It would be beneficial if you could also consider the high-level comments from Reviewer-kGcR and Reviewer-iZKM, as these comments highlight our contributions from a broader perspective. *We believe a paper’s impact should be evaluated from both high-level and detailed viewpoints.*
>
> [Q1] The definition of embedding is confusing.
>
> **Embeddings** is a commonly used term in the context of foundation models. It typically refers to “the transformation of raw data into continuous vector representations that capture the semantic relationships and essential features of the data, enabling efficient and meaningful processing by the model [1]”.
>
> - A well-known early work, “word2vec,” introduced the concept of a continuous vector space for natural language. This continuous vector space is commonly referred to as “the embedding space.”
> - In addition, a recent well-known work, “The Platonic Representation Hypothesis [3],” presented as a position paper at ICML 2024, uses the term “embeddings” extensively, mentioning it 12 times throughout the paper.
>
> [Q2] Confusing in the definition of interleave.
>
> Firstly, thank you for taking the time to understand the term “interleave.” We will provide a more comprehensive explanation here.
>
> - Your understanding “It is not limited to image+text or text+text; it can be image+text+image+text” is correct. This is clearly illustrated in Figure 2 (2) of our main paper.
> - The terminology “interleave” was not introduced by us; it first gained attention within the community through the Flamingo paper [4] by Google, which had a significant impact in 2023. In Fig.7 of Flamingo, it clearly provides an example of an interleaved token “<BOS>Cute pics of my pets!<EOC><image>My puppy sitting in the grass.<EOC><image> My cat looking very dignified.<EOC>”. Hope this can help you to have a better understanding on interleave.
>
> [Q3] The method is somewhat simplistic and lacks innovation, not much new design.
>
> I would suggest that simplicity with minimal overhead is best for model design.
>
> - As stated on the first page of LLaMA3 [5] paper, it indicates:
>
>     *“We believe there are three key levers in the development of high-quality foundation models: data, scale, and **managing complexity**.”*
>
>     Managing simplicity is very important for the potential of success for scaling up.
>
> - Our main contribution in the methods section is the design of a sophisticated interface that leverages the attention mask for various downstream tasks without any overhead. We have provided a comprehensive unification demonstration of all the proposed tasks under the FIND interface in [Common Question 4]. We will give a preview (notations are defined in the PDF) below for fast reference:
>
> [Q4] Why the interleave approach is challenging.
>
> - Interleave retrieval was introduced in the FROMAGe paper [6] without any benchmark, prompting us to propose an interleave retrieval benchmark and use FROMAGe as a baseline for comparison.
> - For interleave retrieval, we compared FIND with strong baselines like ImageBIND and BLIPv2, which lack interleave understanding, and FROMAGe, which does. Table.3 demonstrates that FIND achieves the best performance in interleave retrieval.
> - We are the first to propose interleave grounding, a challenging task because it requires joint reasoning of vision and language information. As shown in Table 3, separating grounding vision and language tokens (SEEM baseline) results in significantly worse performance on interleave grounding compared to FIND, with differences as high as ~10 points across various scales and datasets.
>
> [Q5] Why not just treat vision and language tokens as a sequence?
>
> - Most interleave methods treat vision and language information sequentially. However, pixel-level information may require multiple tokens (e.g. 512 tokens) for accurate representation. To address this, we separate queries and tokens, allowing us to compress token information into queries for interleave retrieval and segmentation.
>
> [1] GPT4-o with prompt “what does embedding means in the context of foundation model in short”.
>
> [2] Mikolov, T., Chen, K., Corrado, G., & Dean, J. (2013). Efficient estimation of word representations in vector space. ICLR 2023.
>
> [3] Huh, M., Cheung, B., Wang, T., & Isola, P. (2024). The platonic representation hypothesis. ICML 2024.
>
> [4] Alayrac, Jean-Baptiste, et al. "Flamingo: a visual language model for few-shot learning." NeurIPS 2023.
>
> [5] Dubey, Abhimanyu, et al. "The Llama 3 Herd of Models." *arXiv preprint arXiv:2407.21783* (2024)
>
> [6] Koh, Jing Yu, Ruslan Salakhutdinov, and Daniel Fried. "Grounding language models to images for multimodal inputs and outputs." ICML 2023.

---

> > ### Author Response · Authors · 2024-08-10
> >
> > Dear Reviewer kpHd,
> >
> > We sincerely appreciate your thorough review of our paper and the valuable feedback you have provided. We would be grateful if you could review our rebuttal and share any additional feedback or questions you might have. If the rebuttal clarifies any points of confusion or if the comments from other reviewers (such as iZKM and AFHH) offer new insights into the paper, we kindly ask you to consider re-evaluating your rating. Thank you once again for your time and efforts!
> >
> > Best,
> > Authors.

---

> ### Comment · Reviewer_kpHd · 2024-08-13
>
> Thanks very much for the author's response. Regarding the question about Q3, I think the author misunderstood my point. I agree that the simpler the design of network methods, the better. The academic work that involves additional design for improving performance is just a paper; the industry would never adopt complicated methods, and in this regard, I believe I am on the same view as the author. However, simplicity does not mean a lack of contribution, or perhaps the way the author presents their ideas makes it difficult for me to grasp the key points of contribution.
>
> Figure 4 is particularly difficult to understand.
>
> If the other reviewers find it easy to understand, AC can disregard my opinion. :)
> I apologize, but I still prefer to maintain my original score.

---

> ### Author Response · Authors · 2024-08-13
>
> Thanks so much for the reviewer's response : ) We are actually on the same page for the simplicity of the design choice, especially when the reviewer gives the following comments:
>
> *“I agree that the simpler the design of network methods, the better. The academic work that involves additional design for improving performance is just a paper”*
>
> The reviewer actually gives comments in a neutral and constructive way. But we the authors want to emphasize, that our work is not lacking in contributions; **it represents a highly efficient and effective unification of foundation model embeddings across both granularity and modality, enabling seamless communication.** Achieving this unification is a significant, non-trivial effort. The authors focused on integrating all components with minimal overhead, which may make the individual contributions appear limited. Although we prepared similar explanations for R4-AFHH, who also found the paper challenging to understand, ***we want to re-emphasize the content here to endeavor the reviewer’s support***.
>
> > **Impact and Novelty**
> >
>
> On the road towards the most powerful “GPT4-V”, there are three major contents for the final models:
>
> 1. Tokenization across modality.
> 2. Effective communication and reasoning between modalities.
> 3. Detokenized to human language.
>
> Our model focuses on the effective communication and reasoning between modalities (**Point 2**), where these components should work together seamlessly. In the multimodal section (7.2) of LLaMA 3.1 [1]. their approach is similar to ours in terms of design choices, but we extend this concept to a finer granularity (pixel-image, phrase-paragraph). **Our approach is novel in this direction, addressing an area that is essential yet underexplored**.
>
> For approaches focusing on **instructional tuning** (e.g., LLaVA), **they primarily work on Point 3**, detokenizing into human language for steerable interaction. *We emphasize that our exploration is a parallel effort, addressing different aspects of the problem.*
>
> > **High-Level Motivation**
> >
> 1. We aim to align embeddings across different granularities (e.g., segments and images for visual understanding; paragraphs and phrases for language) within foundation models like X-Decoder and LLaMA.
> 2. We seek to align embeddings between foundation models across modalities (e.g., Vision and Language), consistent with the Platonic representation hypothesis.
> 3. We aim for the aligned embeddings to communicate seamlessly and efficiently.
> 4. We want to identify the optimal design choices for embedding extraction from different foundation models.
> 5. We believe all understanding tasks fall under the broader scope of information mapping.
>
> > **Interface Design aligned with Motivation**
> >
> 1. We design the interface to incorporate queries and prompts that unify granularity within a foundation model. Specifically, queries can attend to an arbitrary number of prompts through content attention, enabling unified granularity embeddings.
> 2. Additionally, since queries can attend to prompts across different modalities through content attention, we align these modalities within the same representation space via queries.
> 3. The condition attention allows queries that span granularity and modality to communicate efficiently.
> 4. We utilize object queries for vision embeddings and exhaustively test which layer embeddings are optimal for language models like LLaMA.
> 5. Projection and similarity mappings enable the unification of all the recognition tasks under a consistent framework.
>
> > **Experiments proves the Design choice and Motivation**
> >
>
> We evaluated generic segmentation, interactive segmentation (Table 2, Main Paper), and text-paragraph grounding (Fig. 1, Second Row, Main Paper) to demonstrate unification across granularity within the foundation model, in both vision and language.
>
> 1. We assessed grounded segmentation and image-text retrieval (Table 2, Main Paper) to validate cross-modality alignment (vision and language). Additionally, we confirmed that this alignment is invariant across foundation models trained independently (Table 4, Main Paper).
> 2. To evaluate the effectiveness of cross-granularity and cross-modality communication, we developed the FIND-Bench for interleave grounding and retrieval. FIND-Bench is compared with other models focused on information fusion in Table 3.
> 3. The design choices for vision have been explored in previous works (e.g., Mask2Former, X-Decoder, SAM). We conducted an ablation study on language design choices in Table 5 (Main Paper, Lower Part) to determine how to extract the correct information from language models.
> 4. In Table 5 (Upper Part), by gradually removing each task in FIND, we demonstrated the effectiveness of similarity mapping on recognition tasks.
>
> [1] The Llama 3 Herd of Models

---

> > ### Comment · Reviewer_kpHd · 2024-08-14
> >
> > Thanks for the author's reply, which has given me a better understanding of the paper's motivation.
> >
> > To be honest, the writing of the paper needs significant improvement (although this shouldn't be the primary criterion for judging whether the paper should be accepted). I hope the author will improve the writing in future versions of the paper.
> >
> > Additionally, regarding the method in the paper, for example, in Figure 1, there are multiple language encoders—why are multiple language encoders necessary? Isn't one sufficient? Is it a good choice to make the model redundant in order to achieve a slight performance improvement? Having multiple task decoders is fine because the tasks are varied.
> >
> > After reading the author's rebuttal, I have raised my score. However, I still hope the author will improve the writing in the final version, especially focusing on the motivation and related aspects.

---

> > > ### Author Response · Authors · 2024-08-14
> > >
> > > Thanks for the reviewer's response, it is quite encouraging. And we again thank you for your suggestion on the gradient for paper improvement.
> > >
> > > [Question] Multiple language encoders in Figure 1.
> > > If reading the figure zoomed in or in detail, there is only **one solid black arrow**, which is the language encoder that we are using. We just show the potential of our model that can integrated with **one of** the language encoders or vision encoders.
> > >
> > > Again, thanks for the understanding efforts in the rebuttal session, we will improve the camera-ready version if any.

---

> > > > ### Comment · Reviewer_kpHd · 2024-08-14
> > > >
> > > > Thank you for the prompt explanation. I just feel that there's no need to even consider the situation with multiple language decoders here, and Figure 1 can easily cause misunderstandings, leading readers to mistakenly think that your model selects one from multiple language encoders (if that's indeed how it works). Of course, this is just a minor issue; the main issue is still the writing of the paper, which makes it difficult for readers to grasp the motivation behind the work.
> > > >
> > > > In any case, I hope the author can improve the writing in the revision, so that the community can gain more insights rather than just publishing a paper.  Wish the author all the best. :)

---

> > > > > ### Author Response · Authors · 2024-08-14
> > > > >
> > > > > Thanks for sharing the information Figure 1 is easy to be misunderstood, we will improve the writing according to the reviewer's comments for sure. Lastly, thanks for your encouragement on "the community can gain more insights rather than just publishing a paper".

---

### Official Review · Reviewer_kGcR · 2024-07-27

**Soundness:** 3
**Presentation:** 3
**Contribution:** 4
**Rating:** 7
**Confidence:** 5

**Summary:**

The paper explores a unified multimodal embedding space across three image-text interleaved tasks, covering different granularity levels from image-level to pixel-level tasks.

**Strengths:**

1. The work investigates various multimodal tasks under image-text interleaved inputs, including grounding, retrieval, and segmentation, providing rich semantic understanding from image-level to pixel-level. The exploration of a unified embedding space is meaningful.
2. The paper proposes a new benchmark, FIND-Bench, which includes new training and evaluation ground truths for interleaved segmentation and retrieval.
3. The writing and visualizations are clear and well-presented.

**Weaknesses:**

1. It would be valuable to explore the effectiveness of the proposed method on larger datasets.
2. The authors also mention recent multimodal models such as Llava and BLIP-v2. It would be interesting to discuss how the current approach compares with these recent methods, as they also aim for modality unification. More insights into the differences between these approaches and whether the current method achieves similar results with less data or solves problems that the aforementioned models do not address, such as pixel-level visual tasks, would be helpful.

**Questions:**

Please see the weakness.

**Limitations:**

Yes.

---

> ### Author Rebuttal · Authors · 2024-08-07
>
> We greatly appreciate your clear and insightful comments. We have made our best effort to address them carefully below and in [Common Question 1], and [Common Question 3]:
>
> [Q1] Experiment on a Larger dataset.
>
> Thank you so much for your interest in scaling up the training recipe. Unfortunately, we do not have sufficient computational resources at the moment. However, we have proposed a new dataset in FIND-Bench, incorporating the latest SAMv2 dataset for interleaved grounding in video frames, to enable further benchmarking. We hope this new dataset will unlock more capabilities of our FIND interface. We are open to future collaborations on scaling up.
>
> → SAMv2 Benchmark statistics.
>
> | Dataset | # Images (videos) | # Instances | Avg Text Length |
> | --- | --- | --- | --- |
> | sav_val | 136 | 532 | 221 |
> | sav_test | 116 | 451 | 209 |
>
> → SAMv2 Experiment results.
>
> |  | sav_val |  |  | sav_test |  |  |  |
> | --- | --- | --- | --- | --- | --- | --- | --- |
> | Model | mIoU | cIoU | AP50 | mIoU | cIoU | AP50 | Latency |
> | FIND_Focal-T_LLaMA_640 | 58.9 | 63.7 | 65.6 | 57.8 | 59.4 | 66.5 | 0.0809 s/iter |
> | FIND_Focal-T_UniCL_640 | 58.3 | 65.7 | 65.9 | 60.1 | 60.3 | 68.7 | 0.0528 s/iter |
> | FIND_Davit-d5_LLaMA_640 | 62.5 | 69.8 | 71.6 | 60.2 | 64.6 | 71.3 | 0.1978 s/iter |
> | FIND_Davit-d5_UniCL_640 | 61.7 | 66.0 | 72.3 | 62.0 | 65.6 | 72.2 | 0.1699 s/iter |
> | SoM + GPT4o | 75.3 | 83.6 | 75.3 | 76.7 | 81.2 | 76.7 | ~10 s/iter |
>
> We also compare our model with the current most capable model GPT4o with SoM labeled mark, the marks are computed by SEEM with most capable vision backbone.
>
> [Q2] Difference and benefits in addition to LLaVA and BLIPv2.
>
> This is a very good question that deserves the attention of all reviewers. We have compared our work with LLaVA and BLIPv2 in [Common Question 1] and discussed the importance of mapping-based methods in the era of steerable models in [Common Question 3].

---

> > ### Comment · Reviewer_kGcR · 2024-08-10
> >
> > After reading the rebuttal, I will maintain my positive score.

---

> > > ### Author Response · Authors · 2024-08-10
> > >
> > > Thanks for your positive comments : )

---

### Official Review · Reviewer_iZKM · 2024-07-29

**Soundness:** 3
**Presentation:** 3
**Contribution:** 3
**Rating:** 6
**Confidence:** 3

**Summary:**

The authors propose a benchmark for evaluation of what they call 'interleave understanding', or tasks which depend on the embeddings which are aligned across both modalities and task granularity, and they call this benchmark FIND-Bench. Find-Bench includes variants of segmentation and retrieval tasks, derived mostly from COCO datasets. The authors then propose a method for lightweight  fusion of llm and vision features with a multi-task objective, and find the model to be effective at various tasks.

**Strengths:**

Originality: The idea of making a universal benchmark for grounding, retrieval, and segmentation at various granularities is  novel. The solution of aligning existing foundation models with light-weight multi-task tuning seems to both be an effective solution and in line with the underlying motivation for foundation models.

Quality: The model design is reasonable, and the experiments are relatively extensive.  The ablations are interesting, especially the feature embeding layer for the llm.

Clarity: The motivation and findings, in addition to most of the technical details, of the paper are clearly presented. See weaknesses for a few cases where more exposition is needed.

Significance: The ability for a model to become a universal model for vision tasks is important, and the benchmark does test a form of universality. It would be helpful to further strenghen this point with motivation from real-world use-cases. See weaknesses.

**Weaknesses:**

Missing related works: The problem of Composed Image Retrieval e.g. ([1]) is quite related to interleaved retrieval. The authors should discuss the differences between the proposed work and the existing Composed Image Retrieval works.

Missing Method Description: Specifics of the sampler, especially for text, are missing in the main paper.  In fact, it is unclear what the 'Embedding Sampler' in Figure 4(b) does.

Missing Baselines: Although the number of baselines is relatively large, it does seem like LLaVA-type models (eg. [2]) can handle interleaved image-text inputs. Is it possible to retrieve with them?

Improved Clarity: In some cases, clarity needs to be improved. In addition to some missing method description, there is also a missing task description of Interactive Segmentation in section 3.1.1.

Motivation: Although the benchmark proposed is interesting, there is a limited connection to what capabilities it has the ability to unlock. Benchmarks, in order to be very impactful, should somehow be connected to an ability models should have

[1].Saito, Kuniaki, et al. "Pic2word: Mapping pictures to words for zero-shot composed image retrieval." Proceedings of the IEEE/CVF Conference on Computer Vision and Pattern Recognition. 2023.

[2] Jiang, Dongfu, et al. "Mantis: Interleaved multi-image instruction tuning." arXiv preprint arXiv:2405.01483 (2024).

**Questions:**

See weaknesses. Also, the FIND pipeline uses unimodal LLM and Vision Encoder, and unifies them with some training. There are many multimodal encoders these days like LLaVA[3], why not use those?

To focus the rebuttal, from my end, additional experiments supporting the weakness of missing baselines are nice-to-have but improved text clarity, related works, discussion of method is more important to me.

[3] Liu, Haotian, et al. "Visual instruction tuning." Advances in neural information processing systems 36 (2024).

**Limitations:**

Limitations are discussed in minimal way, but I see no major societal impacts.

---

> ### Author Rebuttal · Authors · 2024-08-07
>
> Thank you very much for your comprehensive reviews. We are motivated to address your concerns in detail below. We hope this clarifies your questions and enables you to have a better impression of our paper.
>
> [Q1] Missing related work with “Pic2word”.
>
> Thanks for pointing out this related work, this composed image retrieval is very related, we have evaluated their method on demo examples shown in the **rebuttal PDF**, section Compared with Pic2Word Baseline.
>
> - It clearly shows that FIND has better multimodal reasoning capability, pic2word may tend to favour one of the entities in the query sentence and neglect the reasoning between multiple entities.
> - Meanwhile, FROMAGe is a concurrent work with Pic2work that is much closer to our setting, we have compared interleave retrieval with FROMAGe in Table. 3 and showing privilege results.
> - Additionally, we want to mention that our approach has more capabilities than Pic2work where FIND can do genetic, interactive segmentation, and grounded interleaved segmentation that is not achievable by Pic2Work.
>
> [Q2] Missing details on embedding sampler.
>
> We appreciate your attention to this issue. We address it in the main rebuttal section under **[Common Question 2]**.
>
> [Q3] Missing baselines in comparison with LLaVA or LLaVA types of models (e.g. Mantis).
>
> - We have discussed the relationship with LLaVA in the related work section in L63 - L66. In addition, the evaluation benchmark of LLaVA and FIND is very different because their output type is different. Please refer to **[Common Question 1]** in the main rebuttal for a detailed explanation.
> - LLaVA does not support interleave understanding, LLaVA is multimodal understanding (order of image, text doesn’t matter), in Figure. 2 (2) we clearly compare multimodal and interleave understanding.
> - In addition, **LLaVA only supports one image input, even in LLaVA 1.6**, which does not support any of our tasks as well. As of last month, the LLaVA-Next-Interleave [1] technical report began supporting multiple images as input, which is later than our implementation. However, they still do not support retrieval and segmentation.
> - **Mantis is definitely concurrent work**; their first arXiv version was released on May 2, 2024, making it not a desirable comparison with a NeurIPS submission. Additionally, they focus on language output like QA or captioning, which is not comparable with FIND, as stated in **[Common Question 1]** for LLaVA-type models.
>
> [Q4] Missing Description on Interactive Segmentation.
>
> Thank you for pointing this out. We have not described what interactive segmentation is in FIND. We will add this to the camera-ready version, if applicable. By default, interactive segmentation involves locating the relevant segment in the image with human reference, such as a point, bounding box, or scribbles. SAM [2] and SEEM [3] are two good references for understanding interactive segmentation.
>
> [Q5] Unclear benchmarking capabilities and lack of strong baselines.
>
> - Our FIND-Bench for interleave retrieval and grounding **benchmarks the capabilities of joint vision and language understanding**. For example, in the scenario “A [Image: Dog] is sitting on the [Text: Bench]”, we want the phrase “is sitting on” to be reasoned across both the image and the text contents. Our evaluation metrics will penalize instances where the content cannot be accurately interpreted (e.g. wrong retrieval/segmentation).
> - Actually, at the time of our submission, Grounding-SAM, BLIPv2, and ImageBIND were all very strong baselines, **utilizing industry-level computational resources**. However, we appreciate you mentioning this. We create a new benchmark on the latest SAMv2 dataset, and benchmark it with the most capable multimodal foundation model GPT-4o.
>
> → SAMv2 Benchmark statistics.
>
> | Dataset | # Images (videos) | # Instances | Avg Text Length |
> | --- | --- | --- | --- |
> | sav_val | 136 | 532 | 221 |
> | sav_test | 116 | 451 | 209 |
>
> → SAMv2 Experiment results.
>
> |  | sav_val |  |  | sav_test |  |  |  |
> | --- | --- | --- | --- | --- | --- | --- | --- |
> | Model | mIoU | cIoU | AP50 | mIoU | cIoU | AP50 | Latency |
> | FIND_Focal-T_LLaMA_640 | 58.9 | 63.7 | 65.6 | 57.8 | 59.4 | 66.5 | 0.0809 s/iter |
> | FIND_Focal-T_UniCL_640 | 58.3 | 65.7 | 65.9 | 60.1 | 60.3 | 68.7 | 0.0528 s/iter |
> | FIND_Davit-d5_LLaMA_640 | 62.5 | 69.8 | 71.6 | 60.2 | 64.6 | 71.3 | 0.1978 s/iter |
> | FIND_Davit-d5_UniCL_640 | 61.7 | 66.0 | 72.3 | 62.0 | 65.6 | 72.2 | 0.1699 s/iter |
> | SoM + GPT4o | 75.3 | 83.6 | 75.3 | 76.7 | 81.2 | 76.7 | ~10 s/iter |
>
> - Lastly, we want to emphasize to the reviewer that the benchmark is only one aspect of our contributions. Our main contribution lies in proposing the FIND interface, which leverages the raw foundation model embeddings for various downstream tasks.
>
> [Q6] Why not use LLaVA vision encoder?
>
> - LLaVA does not have its own vision encoder; it uses the pretrained CLIP encoder, which can be found in: https://github.com/haotian-liu/LLaVA/blob/c121f0432da27facab705978f83c4ada465e46fd/llava/model/multimodal_encoder/clip_encoder.py#L7. Moreover, they do not fine-tune the weights, as indicated by the use of torch.no_grad() in the forward path of: https://github.com/haotian-liu/LLaVA/blob/c121f0432da27facab705978f83c4ada465e46fd/llava/model/multimodal_encoder/clip_encoder.py#L133.
>
> [Q7] Rebuttal focus: new baseline numbers (Important), improved text clarity, related works, discussion of method (More important).
>
> Thanks for your clarification, we have summarize the rebuttal in the following contents:
>
> - New Results: Q1, Q5.
> - Discussions: Q2, Q3, Q4, Q6.
>
> [1] Li, Feng, et al. "LLaVA-NeXT-Interleave: Tackling Multi-image, Video, and 3D in Large Multimodal Models." *arXiv preprint arXiv:2407.07895* (2024).
>
> [2] Kirillov, Alexander, et al. "Segment anything." ICCV 2023
>
> [3] Zou, Xueyan, et al. "Segment everything everywhere all at once." NeurIPS 2023.

---

> > ### Comment · Reviewer_iZKM · 2024-08-08
> > **Thank you**
> >
> > I thank the authors for their responses. They have adequately addressed most clarity concerns (e.g. comparison to Pic2Word) and discussion (e.g. details on embedding sampler). Therefore, I raise my score my one point. I strongly encourage all clarifications to make it into any final version. Moreover, because the embedding sampler is identity for the most part, my personal take is that introduces unneeded complexity into the paper writing. I encourage the authors to consider this perspective in any further version.

---

> ### Author Response · Authors · 2024-08-08
> **Thanks for your feedback : )**
>
> Thanks so much for your prompt reply! We are happy that the rebuttal materials solve your confusion, for the embedding sampler, it seems that this creates confusion for nearly all the reviewers, so that we will correct this in the camera-ready version if any. Thanks again for increasing the rating!

---

### Author Rebuttal · Authors · 2024-08-07

→ We thank all the reviewers for their constructive comments with the **following strengths listed**:

**[Novelty (iZKM, kGcR, kpHd, AFHH)]**: The idea of making a universal benchmark for grounding, retrieval, and segmentation at various granularities is novel. The exploration of a unified embedding space is meaningful. The starting point and motivation of the paper is good. The paper idea is novel and promising if extended well.

**[Experiment (iZKM, kpHd, AFHH)]**: The experiments are relatively extensive, and the ablations are interesting. The experimental results are very good. The paper conducts an extensive experiments on COCO and compares with valid baselines.

**[Writing (iZKM, kGcR)]**: Paper is clearly presented for technical details and findings. The writing and visualizations are clear and well-presented.

**[Benchmark (iZKM, kGcR)]**: The paper proposes a new benchmark, FIND-Bench, which includes new training and evaluation ground truths for interleaved segmentation and retrieval. The benchmark does test a form of universality of vision problems.

In general, the reviewers regard the paper with good novelty and strong experimental results.

→ We also addressed the common question raised by the reviewers:

**[Common Question 1]**: Clarification of the main idea and the relationship with instruction-based methods (e.g., **LLaVA**) and Q-Former styled methods (e.g., **BLIPv2**).

- Logically, our method is an experimental proof of “The Platonic Representation Hypothesis [1],” which suggests:

    *“Neural networks, trained with different objectives on different data and modalities, are converging to a shared statistical model of reality in their representation spaces.”*

    Our experiments in Table 4 clearly indicate that although SAM and LLaMA are trained on very different datasets, the embeddings of the two models can be projected into the same embedding space for similar semantic meanings. Additionally, it shows that models trained on very different objectives (segmentation and text generation) can converge to a shared statistical model.

- LLaVA uses a fixed vision model as a tokenizer for LLM to enable generative visual QA. Meanwhile, BLIPv2 aligns the vision output with the language input as well. Both models have tuned the LLM weights, which does not provide clear insight into how vision foundation models and LLMs are connected. Unlike these, FIND aligns the vision backbone features (embeddings) with the intermediate layer features (embeddings) of LLMs, which is distinctly different from LLaVA and BLIPv2 (L62 - L64, Table 5).
- The output modality of LLaVA and BLIPv2 is very different from ours. They use LLMs as generative language decoding tools for QA and captioning, whereas we focus on mapping problems (e.g., retrieval, segmentation, grounding, etc.). **We are not supposed to compare with LLaVA, as it does not support retrieval and segmentation, and we do not support question answering.** For BLIPv2, it supports retrieval, so we compare the retrieval results in Table 3.
- The training budget of LLaVA and BLIPv2 is much heavier than FIND. (1) We only tune the vision projector (6 layers of transformers) and the FIND interface (9 layers of transformers) in a single stage. In contrast, both BLIPv2 and LLaVA involve multistage training and LLM tuning, requiring much higher computation for tuning.

**[Common Question 2]**: **Embedding sampler** is not well defined.

- The embedding sampler is a variant of the visual sampler presented in “Segment Everything Everywhere All at Once [2].” In L173, we mention that “Technically, the embedding sampler is usually an interpolation or grid sample layer in PyTorch.”
- Except for interactive segmentation or interleave segmentation with visual reference (they are visual samplers in [2]), the embedding sampler is an identity grid sampling layer (i.e., it keeps all the embeddings). We maintain the potential for any future sampling for long language context, so by using the term embedding sampler.
- As stated in our main paper:

    *“Technically, the embedding sampler is usually an interpolation or grid sample layer in PyTorch.”*


**[Common Question 3]**: Why understanding and **mapping is important in the steerable era**.

- Understanding the mapping is crucial for comprehending the embedding properties of foundation models.
- Aligning the mapping serves as an intermediate step toward steerable interaction, where instruction-tuned models decode embeddings into human-understandable language.
- As shown in Table 5, vision embeddings have the best alignment with the intermediate embeddings of LLaMA. This finding supports the design choice in the multimodal Llama3, as discussed in “The Llama 3 Herd of Models [3],” which utilizes cross-attention layers to interact with vision embeddings. We believe this design choice is not ideal for both LLaVA and BLIPv2, as noted in L64-66.

[1] Huh, M., Cheung, B., Wang, T., & Isola, P. (2024). The platonic representation hypothesis. ICML 2024.

[2] Zou, Xueyan, et al. "Segment everything everywhere all at once." NeurIPS 2023.

[3] Dubey, Abhimanyu, et al. "The Llama 3 Herd of Models." *arXiv preprint arXiv:2407.21783* (2024)

---

### Author Response · Authors · 2024-08-07

**[Common Question 4]**: Details on **task unification across modality and granularity**.

In Section 3.2.2 (Main Paper), we provided a comprehensive overview of the general pipeline of FIND. Here, we focus on the task-specific interface design choices. The pipeline comprises three main components: (1) Embeddings, which include prompts (p), and queries (q) as introduced in Section 3.2.2 (Main Paper). Prompts are multimodal embeddings containing relevant information, while queries are learnable embeddings that aggregate information from the prompts. For instance, for **image prompts** (a.k.a visual features of an image) we denote them as **p.image**, *other notations in the table below also follow this naming protocol*. ****(2) Operators, which incorporate both content and condition attention, and are responsible for information accumulation and exchange. The arrows ←, ↔ denote the attention direction. (3) Projection, which maps the queries into semantic or pixel space. The table below shows details of all task-specific design choices for the FIND interface, including embeddings, operators, and projection.

| Task | Prompts | Embeddings | Content Attention | Condition Attention | Projection |
| --- | --- | --- | --- | --- | --- |
| Generic Segmentation | image, class | object, class | q.object ← p.image | p.* ↔ p.*, q.* ↔ q.* | Pixel, Semantic |
| Grounded Segmentation | image, text | grounding, text | q.grounding ← p.image, q.text ← p.text | p.* ↔ p.*, q.* ↔ q.* | Pixel, Semantic |
| Image-Text Retrieval | image, caption | image, caption | q.image ← p.image, q.caption ← p.caption | q.grounding ← p.text | Semantic |
| Interactive Segmentation | image, spatial | segment, spatial | q.segment ← p.image, q.spatial ← p.spatial | p.* ↔ p.*, q.* ↔ q.* | Pixel, Semantic |
| Interleave Grounding | image, interleave | entity, interleave | q.entity ← p.image, q.interleave ← p.interleave | q.entity ← p.interleave | Pixel, Semantic |
| Interleave Retrieval | image, interleave | image, _interleave | q.image ← p.image, q._interleave ← p.interleave | p.* ↔ p.*, q.* ↔ q.* | Semantic |

*Table 1. Task-specific FIND Interface. We define each task under the prototype of the FIND interface that enables a shared embedding space, and a unified and flexible architecture for future tasks. Where p, q stands for prompts, queries, and arrows stand for attention direction.*

A comprehensive case study on Interleave Grounding is in the **rebuttal PDF**.

---

### Decision · Program_Chairs · 2024-09-25

**Decision:**

Accept (poster)

**Comment:**

This paper was reviewed by several experts where a plurality argued for accepting the paper.  The proposed dataset and approach seemed generally accepted by reviewers.  Some minor concerns still remain in terms of the positioning of the paper and the significance of the results.  For example, the authors argue that they only support a subset of tasks, but that seems to be contrary to the general purpose of a "foundation model" by only focusing on some tasks.  While this can be argued as common definitions, it is imprecise and should be discouraged.  The approach itself can also be seen as essentially combining multiple models, but this also has other alternatives, such as model soups.  While this wouldn't add more task flexibility, it does result in a more effective model for the tasks it does support.  These weaknesses and others noted by reviewers call into question the significance of the work, but not enough to argue for overturning the plurality recommendation by reviewers.